# Transient power-law behaviour following induction distinguishes between competing models of stochastic gene expression

Andrew G. Nicoll [1], Juraj Szavits-Nossan [1], Martin R. Evans[2] & Ramon Grima [1] ✉

What features of transcription can be learnt by fitting mathematical models of gene expression to mRNA count data? Given a suite of models, fitting to data selects an optimal one, thus identifying a probable transcriptional mechanism. Whilst attractive, the utility of this methodology remains unclear. Here, we sample steady-state, single-cell mRNA count distributions from parameters in the physiological range, and show they cannot be used to confidently estimate the number of inactive gene states, i.e. the number of rate-limiting steps in transcriptional initiation. Distributions from over 99% of the parameter space generated using models with 2, 3, or 4 inactive states can be well fit by one with a single inactive state. However, we show that for many minutes following induction, eukaryotic cells show an increase in the mean mRNA count that obeys a power law whose exponent equals the sum of the number of states visited from the initial inactive to the active state and the number of rate-limiting post-transcriptional processing steps. Our study shows that estimation of the exponent from eukaryotic data can be sufficient to determine a lower bound on the total number of regulatory steps in transcription initiation, splicing, and nuclear export.

A broad molecular-level picture of transcription in eukaryotes is well established[1–3]. This picture involves the opening of chromatin which allows RNA polymerase II (RNAP) to access the promoter region, the sequential binding of several transcriptional factors and of RNAP to the promoter leading to the closed preinitiation complex (PIC), the unwinding of promoter DNA to create an open PIC that starts productive elongation of nascent RNA, and finally the detachment of the RNAP leading to a mature RNA molecule. Despite this detailed molecular knowledge, it has thus far proved difficult to understand the large degree of heterogeneity in gene expression observed in a population of cells[4–6].

Mathematical modelling provides a potential approach to understanding how the observed statistics of gene expression emerge from the molecular description of transcription[7]. While in principle, models can be constructed that encapsulate each and every

biochemical reaction step that is relevant to transcription, in practice this strategy is futile, because given the huge parameter space, its numerical exploration via simulations would require prohibitive amounts of computational power. A practical way forward is through the use of much simpler models composed of a (relatively) small set of effective reactions, where each reaction captures the overall effect of a large number of elementary reactions[8]. These reactions often correspond to the rate-limiting steps (the slowest steps of the mechanism) since these are principal determinants of a model's output.

A large variety of such minimal models of stochastic gene expression has been developed[9,10]. Once a model is formulated, its output can be fit to experimental data, thus leading to the estimation of the rate constants[11–13]. Moreover, by assessing the relative goodness of fit between a suite of models, one can potentially arrive at a best model that captures salient features of the data[14–16]. This model

[1]School of Biological Sciences, University of Edinburgh, Edinburgh, United Kingdom. [2]School of Physics and Astronomy, University of Edinburgh, Edinburgh, United Kingdom. ✉e-mail: ramon.grima@ed.ac.uk

selection holds great promise since in principle it can be used to sift between competing mechanisms that at first sight appear equally likely to explain a set of measurements.

The majority of the published minimal stochastic models of transcription are either special cases of, or can be reduced to, the following *N*-state model:

$$G_1 \xrightarrow{k_1} G_2 \xrightarrow{k_2} G_3 \xrightarrow{k_3} \dots \xrightarrow{k_{N-2}} G_{N-1} \xrightarrow{k_{N-1}} G_N \xrightarrow{k_N} G_1,$$
$$G_N \xrightarrow{\rho} G_N + M,$$
$$M \xrightarrow{d} \emptyset. \quad (1)$$

Here $G_1$ to $G_{N-1}$ (for $N > 1$) are inactive states from which mRNA (*M*) synthesis is not possible and $G_N$ is an active state from which it is. The rate (constant) parameters governing the transition from gene state $i$ to $i+1$, mRNA synthesis in gene state $N$ and mRNA degradation are $k_i$, $\rho$, and $d$, respectively. Each of the various reactions represents an important rate-limiting step in the transcriptional process. While an extension of this model to include reversible reactions between gene states or transcription from multiple gene states is likely more realistic[17–19] (this will be considered later), nevertheless it has many more rate parameters than the irreversible model and hence the latter is in more common usage[20,21]. It can be shown that the irreversible model corresponding to reaction scheme (1) predicts the Fano factor (variance divided by the mean) of mRNA counts to be greater than or equal to 1, which is the case for most genes; however, this means that it cannot describe the sub-Poissonian expression (Fano factor less than 1) measured for a few eukaryotic and prokaryotic genes[16,22]. The case $N = 1$, i.e. no inactive states, is that of constitutive (non-regulated) expression. Bursty expression, i.e. where transcription occurs only in short bursts, separated by intervals of inactivity implies the presence of at least one inactive state, i.e. $N \geq 2$. For example, the commonly used telegraph model, also called the 2-state model, is a special case of this model with $N = 2$ (one inactive state and one active state)[11–13,23–26]. Many genes seemingly exhibit bursty expression[27]. For these, there is evidence that generally the number of active states is just one (since the distribution of the total time spent in the active states is measured to be exponential[20,28]) but the number of inactive states can be greater than one (since the distribution of the total time spent in the inactive states can be non-monotonic with a peak at a non-zero value[20,29]). Most experimental evidence suggests that the number of inactive states is just one for prokaryotic genes[15]. On the other hand, for eukaryotic genes, some studies make the case that it is larger than one[14,15,20,21,29–31] while others are consistent with one[12,25,28]. In particular the telegraph model, with one active and one inactive state, well fits the statistics of mRNA counts for thousands of sequenced mammalian genes[25], seemingly contradicting studies that make a case for models with more inactive states. We emphasize that because the number of inactive states can be interpreted as the number of rate-limiting steps in initiation, their determination is of biological importance since these are often the likely points of gene regulation.

Unfortunately, commonly used methods to determine the number of rate-limiting steps present several challenges. These methods can be divided according to the data used: (i) direct or indirect measurements of the distribution of the total time spent in the inactive states[20,28,31–33]; (ii) measurements of the time-dependent distribution of mRNA counts following a perturbation[14,15]. Methods of type (i) are the most reliable because if the distribution of time in the inactive state is measured to be non-exponential then the number of active states is greater than one. However, direct measurements of this type are challenging because live-cell measurements are low throughput and require genome-editing[34,35]. Methods of type (ii) while indirect are often the preferred choice since mRNA count data per cell can nowadays be easily obtained for several genes using single-molecule fluorescence in situ hybridization (smFISH)[36] or single-cell sequencing (scRNA-Seq)[25]. These require

measurements of the distribution of mRNA counts at several time points, each of which is calculated from an independent sub-population of cells. Inference methods using this type of data maximize the likelihood[14,37] or utilise a Bayesian framework[15] to fit stochastic models of gene expression with varying number of inactive states to temporal count data, in the process selecting an optimal, best-fit model. However, it is currently unclear how robustly these methods perform in determining the number of inactive states in steady-state conditions. There is some evidence from comparison of the telegraph and the *N*-state model with $N = 3$ that it is not easy to estimate the number of gene states from steady-state count data[38,39] but a systematic analysis, especially one for larger values of *N*, remains missing.

These methods also suffer from some notable disadvantages: they are not easy to use because of their technical sophistication and the quality of inference is highly dependent on how well one can correct the data for static extrinsic noise, i.e. cell-to-cell variability in the rate parameters. The latter is a particularly significant issue because static extrinsic noise tends to increase the variance of the distribution of mRNA counts (relative to its mean) and hence can lead to the incorrect estimation of rate parameters[40] or even the number of inactive states[41].

Summarising, we have identified two open questions: (i) is steady-state count data sufficient to estimate the number of inactive states? (ii) can a simple method be used to easily estimate the number of inactive states—one that does not need large volumes of temporal data and is robust to static extrinsic noise? A schematic summarising these questions and the whole model deduction process is illustrated in Fig. 1.

In this paper, we provide answers to these two questions. We consider the model (1) with one active state and a general number of inactive states, i.e. $N \geq 2$. We first conclusively show that steady-state mRNA count data is not typically sufficient to distinguish between the $N = 2$ model[23] and models with $N = 3$, 4 and 5 gene states. In particular, we identify a mapping of rate parameters between the telegraph model and the other types of models that lead to distributions of mRNA counts that in the vast majority of cases are practically model-independent. Next, we devise a simple method to overcome the difficulties of the inference method based on steady-state data and other methods in the literature that utilise the time-dependent distribution of mRNA counts. We analytically show that following induction, the initial increase of the mean mRNA count with time is described by a power law whose exponent is solely dependent on the number of gene states between the initial inactive state and the active state. We confirm the accuracy of the theoretical result using simulations and then apply it to data collected for various cell types to provide a robust estimate of a lower bound for the total number of inactive gene states. Interestingly, our results show that this bound: is not affected by static extrinsic noise due to cell-to-cell variability; is valid for a large class of networks of gene states (of which (1) is a special case); and in principle is sufficient to accurately infer the number of gene states using bulk data, rather than single-cell data.

## Results
### Theoretical limitations of inference from steady-state mRNA count data
We first remark that there are two fundamental limitations to what can be inferred from the steady-state distribution of the number of mRNA counts of the *N*-state model defined by reaction scheme (1). The steady-state distribution of mRNA counts, $P(m)$, is dependent only on the values of $k_i/d$ and $\rho/d$, i.e. the rate parameters normalised by the mRNA degradation rate. Hence in order to achieve a perfect match between the theoretical and an experimentally measured distribution of mRNA counts, it would not be possible to infer the absolute values of $k_i$ and $\rho$ unless the degradation rate $d$ was also estimated by means of a separate experiment. A second limitation is that the steady-state distribution $P(m)$ is invariant to permutations of the set of rate constants $k_1, k_2, \dots, k_{N-1}$, i.e. the rate constants of the reactions from one inactive state to another or to the active state are interchangeable. For

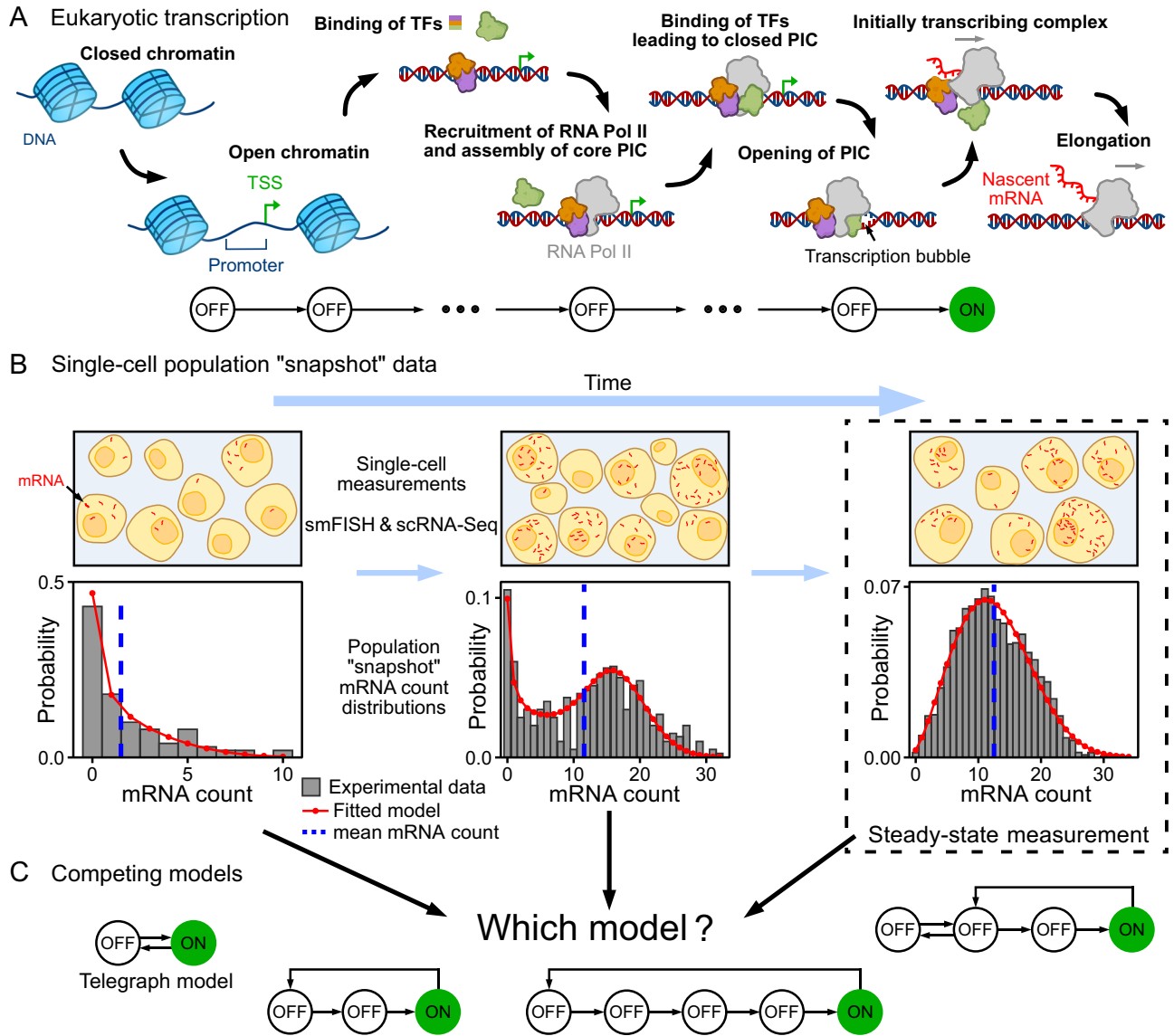

**Fig. 1 | Inference of the number of rate-limiting steps in transcription from mRNA count data using mathematical models of gene expression. A** Schematic of the key stages in eukaryotic transcription. This includes the opening of closed chromatin, exposing the transcription start site (TSS); the binding of various transcription factors (TF) to the promoter; recruitment of RNAP and assembly of the PIC; and successful elongation. From a detailed biological description, it is unclear what are the actual number of rate-limiting steps. **B** Schematic of single-cell population snapshot measurements of the mRNA count in time (including at steady-state) and the corresponding mRNA count distributions. Mathematical models (red lines) can be fitted to experimental data (histograms), obtained using smFISH and scRNA-Seq, with the aim of inferring probable transcriptional mechanisms. **C** It is currently unclear if by fitting steady-state distributions only, the number of rate-limiting steps and the network topology of the true transcriptional mechanism can be identified. For example, can the telegraph model and models with 3, 4, or 5 states be distinguished? Here OFF represents a transcriptionally inactive state and ON an active state. Equally unclear is the success of inference using time-dependent distributions due to static extrinsic noise stemming from cell-to-cell variability. These questions are addressed through an efficient and simple-to-use method using the mean mRNA count (dashed blue line) to overcome the limitations of current methods.

example, this means that by matching of the steady-state mRNA distributions measured experimentally to that of an $N$-state model, we cannot distinguish between a model where there is fast transcription factor binding to the promoter followed by slow RNAP binding from a model where the speeds of these two processes are reversed. See Supplementary Note 1 for a queuing theory-based derivation of the steady-state distribution of mRNA counts of the $N$-state model (see also ref. 42) and a proof of the two aforementioned results.

**A biophysically meaningful mapping between the rate parameters of the telegraph and $N$-state model**

We note that the steady-state distribution of reaction scheme (1) is not invariant with respect to $N$ and hence in principle it might be possible to distinguish between models with different total numbers of states. Of course, there is always the possibility that two models with different $N$ have very similar steady-state distributions of mRNA counts for particular choices of rate parameter values, in which case they would be practically indistinguishable. In particular, since the telegraph model is the commonest model in the literature, it is of interest to understand whether it can accurately match the steady-state mRNA distributions of gene expression models with more than 2 states, i.e. with multiple inactive states.

The natural way to investigate this question would be to (i) generate a large number of rate parameters sets of the $N$-state model; (ii) for each of these sets, use an optimization algorithm to find rate parameters of the telegraph model which minimize the

distance between its steady-state mRNA distributions and that of a given $N$-state model. If this procedure identifies rate parameters for which the minimal distribution distance is very small then it means that the telegraph and the $N$-state model are essentially indistinguishable. While desirable, this procedure is computationally expensive and would be difficult to employ to explore large swathes of parameter space.

To avoid these difficulties we devise a different procedure. We define the distribution $f_N(\tau)$ of the waiting time $\tau$ between two successive transcription events of the $N$-state model, its mean denoted $\langle\tau\rangle_N$ and variance $\sigma^2_{\tau,N}$. The procedure then consists of the following three steps: (i) generate thousands of rate parameter sets of the $N$-state model with $N = 3, 4, 5$; (ii) for each parameter set, calculate the rate parameters of an effective telegraph model by matching its waiting time statistics to that of the $N$-state model. In particular we enforce $f_N(0) = f_2(0)$, $\langle\tau\rangle_N = \langle\tau\rangle_2$ and $\sigma^2_{\tau,N} = \sigma^2_{\tau,2}$. This means that the rate parameters of the effective telegraph model (denoted by a star superscript) are given in terms of the rate parameters of the $N$-state model by:

$$\rho^* = \rho, \quad k_1^* = \frac{1}{\langle\tau_{1\to N}\rangle}\cdot\frac{2}{CV^2_{1\to N}+1}, \quad k_2^* = k_N\cdot\frac{2}{CV^2_{1\to N}+1}, \quad (2)$$

where $\tau_{1\to N}$ is the random time it takes to go from state 1 to state $N$ in the $N$-state model and $CV_{1\to N}$ is its coefficient of variation (ratio of the standard deviation to the mean). For a derivation of Eq. (2) see Methods and for their interpretation see the Discussion section. Finally, step (iii), using the rate parameters calculated in the previous step, calculates the Wasserstein distance (WD) between the cumulative mRNA count distribution of the effective telegraph model ($P^c$) and that of the $N$-state model ($Q^c$) defined as WD $= \sum_i |P_i^c - Q_i^c|$ where $i$ is the number of mRNA counts. If the WD is small over the whole scanned parameter space then we know that generally there exists an effective telegraph model that captures well the salient features of the mRNA count distribution of the $N$-state model. The choice of WD is motivated by its simple interpretation – it considers how much and how far probability mass has to be moved to reshape a histogram into another[43] – and its avoidance of other well-known issues[44].

The theoretical justification of step (ii) is as follows. Using queuing theory, in Supplementary Note 1 we show that the steady-state mRNA count distribution of the $N$-state model is completely determined by the Laplace transform of $f_N(\tau)$, which we shall denote by $f_N^*(s)$, where $s$ is the variable in Laplace space. Now $f_N(0) = \lim_{s\to\infty}[sf_N^*(s)]$, which implies that the condition $f_N(0) = f_2(0)$ forces $f_N^*(s)$ and $f_2^*(s)$ to agree in the limit of large $s$. In addition, by matching the first two moments of the waiting distribution, $\langle\tau\rangle_N = \langle\tau\rangle_2$ and $\sigma^2_{\tau,N} = \sigma^2_{\tau,2}$, we effectively are matching $f_N^*(s)$ and $f_2^*(s)$ for small $s$. This is because by the definition of the Laplace transform the $r$th moment of $f_N(\tau)$ is determined by the $r$th derivative of $f_N^*(s)$ with respect to $s$, hence the series expansion (and Pade approximation) of $f_N^*(s)$ and $f_2^*(s)$ must agree for small $s$. Thus by step (ii), we are guaranteed that $f_N^*(s)$ and $f_2^*(s)$ agree for both small and large $s$ which in turn implies that the steady-state mRNA count distribution of the $N$-state and the telegraph models are also matched to a good degree. Further justifications for step (ii) are that it is very quick to compute since it requires only the evaluation of analytical formulae and that the procedure guarantees that the rate constants of the effective telegraph model are positive and hence physically meaningful (see Methods for a proof). Whilst this is not the only matching procedure that will accomplish this, common alternative matching procedures cannot guarantee this; for example, matching of the first three moments of the waiting time distributions or the first three moments of the mRNA count distributions of the $N$-state and telegraph models can lead to negative rate parameter values.

## Inferring the number of inactive gene states is difficult from steady-state data

In Fig. 2 the steady-state mRNA count distributions of thousands of rate parameter sets for the $N = 3, 4, 5$ state models are compared with those of their effective telegraph models (computed using the previously detailed three-step procedure). Note that to ensure biological relevance, the rate parameter ranges for each model were chosen to comfortably cover the bounds of the telegraph model rate parameters estimated using a range of eukaryotic data in various published studies (see Table 1 of ref. 45 for these estimates and see Supplementary Note 2 for further details on generating the rate parameter sets).

Figure 2A–C shows the probability-probability (P-P) plots comparing cumulative distribution functions of the effective telegraph and the $N$-state models for the 500 rate parameter sets with the largest WD of those sampled. Very similar distributions fall close to the dashed red line (representing identical distributions), the greater the deviations, the more disagreement between the distributions. For each of the $N = 3, 4, 5$ state models, we see minor deviations from this line, indicating broad agreement between the compared distributions. As the number of states increases, we see greater, but not significantly so, deviations as the bulge around this line broadens. Overall these P-P plots show that even for the most statistically distinct distributions, the difference between these distributions is not very large. This suggests that in many cases the telegraph model provides an excellent approximation to the $N$-state models with $N > 2$.

To obtain a deeper insight into the deviations, we categorised the $N = 3, 4, 5$ state rate parameter sets according to the shape and statistical properties of their mRNA count distributions. All sampled distribution shapes fell into the following four groups: shape I—unimodal with Fano factor < 2; shape II—unimodal with Fano factor ≥2; shape III—bimodal with one zero and one non-zero mode; shape IV—bimodal with two non-zero modes. Note that the single mode of both shapes I and II can be either at zero or at a non-zero value. A typical distribution of each shape is seen in Fig. 2D–G, along with the WD between the 5-state and the effective telegraph distribution. A comparison of the WD of these four shapes suggests that the quality of the effective telegraph model fits decreases with increasing shape number, being best for shape I and worst for shape IV.

In Fig. 2H we show by means of box plots that this is indeed the picture that emerges when considering all rate parameter sets sampled: comparing box plots of the same colour, i.e. same number of states, we see that shape I distributions are associated with the lowest median WD, followed by shapes II and III distributions which have similar median WDs that are approximately 5-10 times larger than those of shape I, followed further by shape IV distributions which have median WDs that are 3-20 times larger than of shapes II and III. These trends become less clear when considering not just the medians but also the whiskers of the box plots which shows that there is still considerable variation of the WD within each shape type (Supplementary Fig. 1). In particular for the 3-state model (the commonest $N$-state model with $N > 2$ in the literature) the lower whisker of the box plots extends nearly to $10^{-3}$ for shapes I-III; this suggests that a substantial portion of the parameter space of this model is very well approximated by the effective telegraph model since WDs of this magnitude are essentially indistinguishable by eye (Fig. 2D). Within each of shapes I, II and III, the median WD increases with the number of states of the $N$-state model (compare the three differently coloured box plots for each category). Hence, as expected, the telegraph model fit worsens as the number of gene states increases, though the worst fits (Shape III for the $N = 5$ model), when visually inspected, still manage to capture the main features of the $N$-state distribution (the two peaks and their positions; Fig. 2F). While in principle this suggests that as $N$ increases, the effective telegraph model will eventually become glaringly distinguishable from the $N$-state model, in reality this is unlikely because

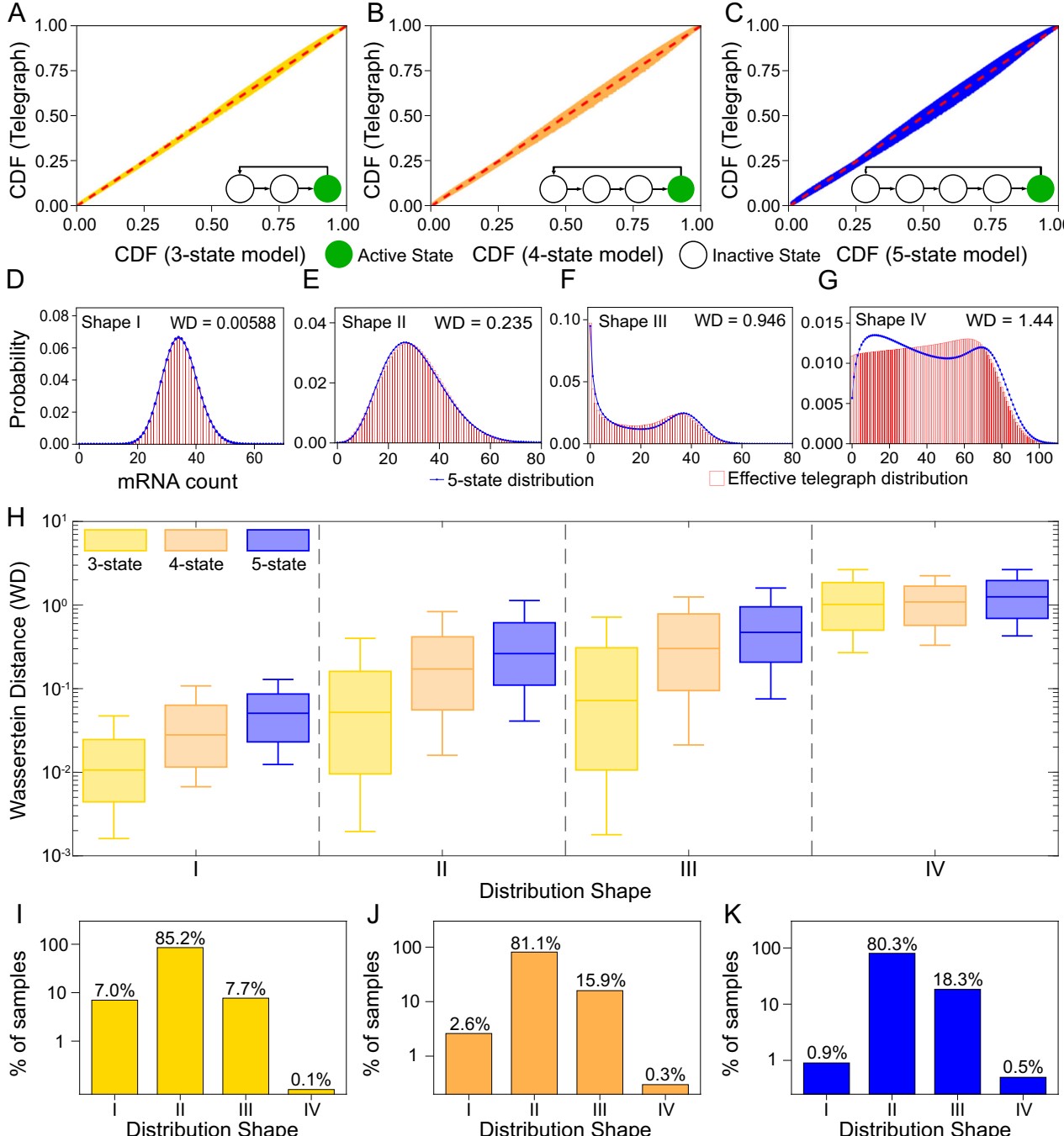

**Fig. 2 | The steady-state mRNA count distribution of the *N*-state model can often be well fit by an effective telegraph model.** Thousands of rate parameter sets of the *N*-state model (*N* = 3, 4, 5) were sampled in the range relevant for eukaryotic gene expression. For each parameter set the mapping in Eq. (2) was used to obtain the rate parameters of an effective telegraph model. **A–C** P-P plots comparing the cumulative distribution function (CDF) of the *N*-state and effective telegraph models. These are constructed from the 500 rate parameter sets with the largest Wasserstein distance (WD) values between the distributions of the two models. Deviations from the line *y* = *x* (red) indicate differences between the distributions. **D–G** Typical steady-state mRNA count distributions of the 5-state (solid blue lines) and effective telegraph models (histograms) for four classes of distributions (Shape I: Unimodal with Fano factor < 2; Shape II: Unimodal with Fano

factor ≥2; Shape III: Bimodal with one peak at zero and another at a non-zero value; Shape IV: Bimodal with peaks at non-zero values. **H** Boxplots displaying summary statistics of the WD between the steady-state distribution of mRNA counts of the *N*-state and effective telegraph model. The horizontal line and the box show the median and the interquartile range, respectively; whiskers extend to the 10th and 90th percentiles; sample sizes given in Supplementary Table 2. **I–K** Bar charts displaying the percentage of each distribution shape sampled across the parameter space. The agreement between the *N*-state and its effective telegraph model becomes worse as *N* increases and is worst for Shape IV because this shape is qualitatively impossible to capture by the telegraph model. See Supplementary Note 2 for implementation details.

until now, the maximum number of states estimated from experimental data is six and this was an isolated case[21].

Therefore, for mRNA count distributions of shapes I-III, our results strongly suggest that the actual number of gene states cannot be reliably inferred from steady-state mRNA count distributions. One possible explanation is that perhaps the log-uniform sampling procedure tended to select parameter sets where one of the $k_i$ values ($i = 1, . . , N − 1$) is particularly small compared to the others; this would imply one rate-limiting step describes gene activation which is consistent with the telegraph model. In Supplementary Fig. 2 we verify that this is not the case: mRNA count distributions from $N$-state models with $N > 2$ and $k_i = a$ for $i = 1, . . , N − 1$ where $a$ is some constant are well fit by an effective telegraph model. This strengthens our earlier conclusion that for shapes I-III, steady-state data is not sufficient to distinguish between $N$-state models with $N = 2, 3, 4$ or $5$.

While our comparison of distributions was done assuming an infinite number of cells, the similarity of the $N$-state and effective telegraph model means that the maximum likelihood of each distribution computed using mRNA count data from each cell in a finite cell population would also be similar. Hence using common model selection criteria such as the Akaike Information Criterion (AIC) (which penalises more strongly models with a larger number of parameters), the telegraph model would often be selected as the optimal one, even if the $N$-state model is the correct one. This intuition has recently been verified in ref. 39 where it was shown that a common model selection procedure applied to simulated count data from the 3-state model incorrectly selected the telegraph model (over the 3-state model) for over 90% of parameter sets if the sample size was 100 cells (typical for scRNA-seq data), and for over 40% of parameter sets for samples with 10, 000 cells (larger than for most smFISH datasets).

Turning our attention to distributions of shape IV; this shape presents an interesting case as it is the only shape out of the four considered for which the telegraph model cannot even qualitatively capture the shape of the distribution of the $N$-state model. As we see for the example in Fig. 2G, the number and position of the modes differ between the 5-state model (blue) and the effective telegraph model (red) - while the former has two modes both at non-zero values, the effective telegraph model has a single, non-zero mode. Now it has been proved that when the steady-state distribution of the telegraph model is bimodal then one of its peaks must lie at zero[46]. Hence clearly for distributions of shape IV, it is impossible for the telegraph model to provide an adequate approximation of the models with $N > 2$. It is also interesting to consider that while slow promoter switching compared to the mRNA degradation rate is a necessary condition for the telegraph model ($k_1/d < 1, k_2/d < 1$) to display bimodal distributions[47], this is not the case for the emergence of bimodal distributions of shape IV in the models with $N > 2$; in particular we found this distribution shape can be obtained when $k_N/d < 1$ and $k_i/d \geq 1$ for $i = 1, ..., N − 1$. Hence we conclude that shape IV distributions are associated with an intermediate switching regime.

Of course, the relevance of our results depends on the actual frequency of the four distribution shapes in the biologically relevant parameter space for eukaryotic transcription. In Fig. 2I–K we see how many of each distribution shape were sampled in the parameter sweep for each $N = 3, 4, 5$ state model. Despite the aforementioned distinction of distributions of shape IV, we actually see that for each $N = 3, 4, 5$ the percentage of shape IV distributions sampled was incredibly low, at 0.1%, 0.3%, and 0.5% respectively. Therefore, even though shape IV distributions are distinguishable from their effective telegraph model distributions in steady-state since they appear so infrequently within the relevant parameter space, we do not expect them to appear commonly from experimental data. Hence our overall conclusion is that the number of gene states is difficult, if not impossible, to reliably estimate from steady-state mRNA count distributions.

## The number of gene states can be estimated from short-time mean count data following induction

Following the difficulty of distinguishing the telegraph and $N$-state models steady-state mRNA count distributions we turn to time-dependent statistics. Inference of the gene state number $N$ from distributions of the mRNA counts measured at several time points has been previously shown using maximum likelihood[14] and Bayesian approaches[15]. In what follows we describe a much simpler method involving only a regression on the mean mRNA count computed for a short interval of time following gene induction.

Consider an experimental setup whereby some gene of interest is inactive before induction by an inducer molecule. From application of the stimulus to the binding of the relevant transcription factor(s) to the DNA, there is a time lag due to translocation of molecules from the cytoplasm to the nucleus and other processes. Hence we shall consider $t = 0$ to be when transcription initiation begins in the nucleus, i.e. we shall assume that the time lag can be experimentally measured and hence its value is known. For simplicity, we shall also assume that the mean mRNA count at $t = 0$ is zero and that in all cells the inactive state of the gene prior to induction is $j$ where $j \in 1, . . , N − 1$. This idealised setup is later relaxed to better capture experimental reality.

We seek to understand what information about the number of gene states $N$ can be deduced from the the moments of the mRNA counts measured a short time after induction. To this end, in Methods, starting from the chemical master equation (CME) for the $N$-state model, we use perturbation theory to show that in the limit of short times, the mean $\langle m(t) \rangle$ of mRNA counts at time $t$ is given by:

$$\langle m(t) \rangle = \rho \frac{\prod_{i=j}^{N-1} k_i}{(N-j+1)!} t^{N-j+1} + O(t^{N-j+2}). \qquad (3)$$

In this equation both time and the rate parameters are non-dimensional since they are rescaled by the mRNA degradation rate parameter, i.e. $t \mapsto td$, $k_i \mapsto k_i/d$ and $\rho \mapsto \rho/d$.

The expression for the mean, Eq. (3), suggests a simple method to extract information about the number of inactive states. For short times after induction, a log-log plot of the mean mRNA count versus time leads to a straight line with slope equal to $N − j + 1$. Since $j = 1, . . , N − 1$, the slope provides a lower bound on the total number of gene states $N$. Specifically the slope estimates the the number of rate-limiting steps between the initial inactive state $j$ and the active state $N$. This result also means that it is difficult to distinguish $N$-state models for which $j = N − 1$ from the 2-state telegraph model because in both cases the measured power law exponent is 2. In Fig. 3A we confirm the derived power law by direct numerical integration of the time-evolution equations for the first moment of the mRNA counts of the $N = 5$-state model−since the system (1) is composed of purely first-order reactions, these equations are simply the deterministic rate equations.

As expected, the power law behaviour is valid only for short times and hence one may question its usefulness in practice. In Methods, we derive a condition for how short the observation time should be so that the $t^{N-j+1}$ scaling law for the mean is valid. As a rule of thumb, the power law for the mean can be considered to be valid over time intervals that are much smaller than the mRNA degradation timescale ($t = 1$ in non-dimensional time units). For mammalian cells, the median half-life of mRNA molecules is many hours, e.g. it is about 9 h for NIH3T3 mouse fibroblasts cells[48], and hence the power law should in this case be observable for many minutes. For prokaryotic cells and simple eukaryotes, the half-lives of mRNA are much less (typically ranging from a few minutes to about an hour and a half for E. coli and budding yeast[49,50]) and hence the power law may be more difficult to observe for these organisms.

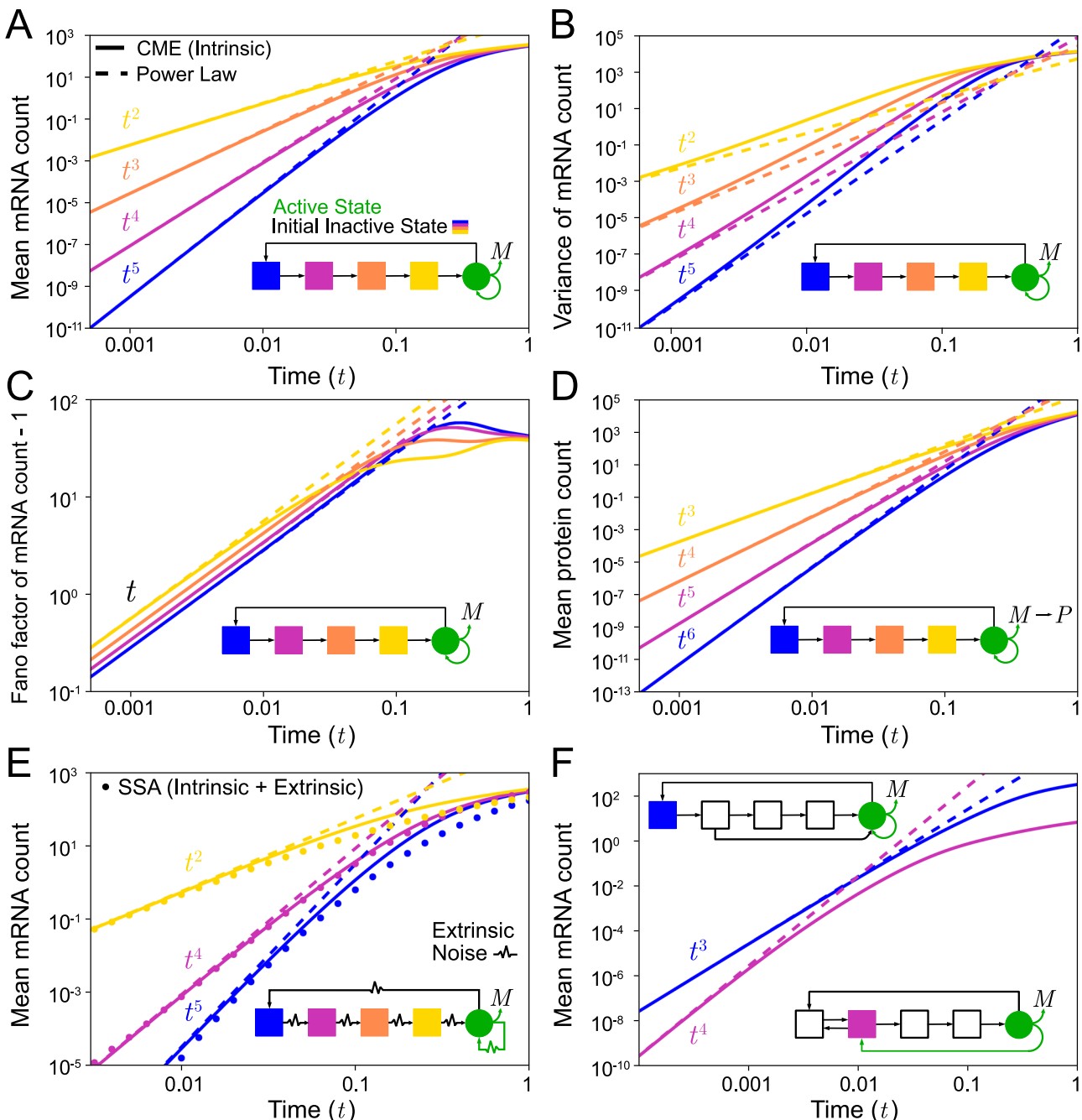

**Fig. 3 | Simulations confirm the predicted power-law behaviour of the time-dependent statistics of the mRNA count following induction. A–C** Mean of mRNA counts, variance of mRNA counts and Fano factor of mRNA counts for the $N = 5$-state model. **D** Mean of mRNA counts for the 5-state model with one extra step modelling the translation of mRNA ($M$) to protein ($P$). **E** Mean of mRNA counts for the 5-state model with cell-to-cell variation of rate parameters (static extrinsic noise); each rate parameter is drawn from a gamma distribution. **F** Mean mRNA count from extensions of the 5-state model that incorporate reversible reactions between gene states and state change upon mRNA synthesis; these are special cases of the system (5). The theory is shown by the dashed lines and simulations by solid lines (obtained by direct numerical integration of the moment equations of the chemical master equation (CME)). In (**E**) the simulations are done in two ways: solid lines are obtained by direct numerical integration of the moment equations of the

CME with rate parameters fixed to the average of the gamma distributions and dots are from stochastic simulations ($10^5$ trajectories) with rate parameters randomly drawn from these distributions. In all cases, there is excellent agreement between theory and simulations for short times. All panels use a log-log scale hence a power-law relationship is represented by a straight line with gradient corresponding to the exponent of the power law. The colours: blue, pink, orange, and yellow, represent the initial inactive gene state after induction: $j = 1, 2, 3, 4$ respectively and are illustrated by the model diagram in each panel. The outward-pointing green arrow represents mRNA synthesis and the other shows the state after synthesis. The parameters were chosen to be within the range relevant for eukaryotic gene expression (the range used for Fig. 2). See Supplementary Note 2 for reaction network schemes.

In Supplementary Note 3 we extend the analytical results to the variance $\sigma_m^2$ and Fano factor $FF$ of the mRNA counts:

$$\sigma_m^2(t) = \rho \frac{\prod_{i=j}^{N-1} k_i}{(N-j+1)!} t^{N-j+1} + O(t^{N-j+2}), FF_m(t) = \frac{\sigma_m^2(t)}{\langle m(t) \rangle} = 1 + \frac{2\rho}{N-j+2} t + O(t^2).$$

$$(4)$$

Note that the short-time power law for the variance is precisely the same as that for the mean. In Fig. 3B and C we confirm this for short times by comparison with the results obtained by direct numerical integration of the time-evolution equations for the first and second moment of mRNA numbers obtained from the CME. This confirms that a log-log plot of the variance of mRNA counts versus time does not provide more information than the mean versus time does and that from the exponent of the Fano factor minus 1 scaling law, it is not possible to obtain any information on the number of states. By comparing Fig. 3A and B, we also note that the true time-evolution of the variance of mRNA counts deviates quicker from the short-time power law than that for the mean of mRNA counts—this is a general property of Markovian systems[51]. This implies that practically the method for extracting the number of states from short-time data is likely only useful when the mean counts are used.

We note that our results also point to a simple method by which some of the rate parameters of the $N$-state model can be estimated. In particular, from simultaneous short-time measurements of the mean and the Fano factor, it is possible to estimate the transcription rate $\rho$ and the product of rate constants $\prod_{i=j}^{N-1} k_i$. The procedure is to first estimate $N - j + 1$ from the slope of a log-log plot of the mean versus time (or from non-linear regression fit of a power law to the data). Using this value and the slope of the FF versus time graph, we obtain $\rho$. Finally given the latter and the earlier estimate of $N - j + 1$, the product of rate constants is obtained from the intercept of the log-log plot of the mean versus time.

We note that power laws similar to Eq. (3) have been previously reported for the short-time behaviour of the distribution of the first-passage time from an initial state to a target state[52–54]. Experiments have shown that this power-law regime encodes information about the number of intermediate states associated with colloidal dynamics in confinement, transport through a biological pore, and the folding kinetics of DNA hairpins[55]. However, this method is difficult to apply to gene expression as direct measurement of the first-passage time distribution from induction to the first transcription event is not straightforward. In contrast, our method based on mean mRNA measurements is practical since these are routinely done using smFISH, single-cell and bulk sequencing.

### Extension of the short-time power law to models including post-transcriptional processing, cell-to-cell parameter variability, and complex gene state networks

Post-transcriptional processing of mRNA may affect the power law exponent. It is simple to show that if splicing and nuclear export can each effectively be described by a single first-order reaction then the short-time exponent deduced from unspliced nuclear mRNA is one less and two less than the exponents deduced from spliced nuclear mRNA data and cytoplasmic mRNA, respectively. Similarly, if we interpret the mRNA species ($M$) in the $N$-state model as the cytoplasmic mRNA, a first-order processing step to describe its translation to protein would imply that the short-time exponent for the latter is one more than that for the cytoplasmic mRNA (see 'Methods')—verified in Fig. 3A and D. On the other hand, if splicing and export can be described by fixed-delay reactions then there is no difference in the exponents deduced from different types of mRNA. If count data is available for mRNA at different stages of its lifecycle then a comparison of the exponents of the short-time power laws for each stage can reveal information about the effective reactions describing processing.

Realistically, the rate parameters of the $N$-state model will generally vary from one cell to another one, a phenomenon called static extrinsic noise[40,56]. This means that the mean mRNA measured in an experiment is in reality described by $\langle m(t) \rangle_e = \int \langle m(t) \rangle P(k_1, \ldots, k_n, \rho) dk_1 \ldots dk_N d\rho$ where $P(k_1, \ldots, k_n, \rho)$ is the joint distribution of all rate parameters in the $N$-state model. Given the solution Eq. (3) it immediately follows that while static extrinsic noise changes the prefactor in front of the power law it does not change the exponent and hence the lower bound of $N$ estimated from a regression of the mean versus time is insensitive to static extrinsic noise. In Fig. 3E we confirm using stochastic simulations the robustness of the short-time power law to static extrinsic noise. Here we use the stochastic simulation algorithm (SSA) to simulate mRNA count data in $10^6$ cells, assuming that the time-independent rate parameters for each cell were drawn from independent gamma distributions. This distribution is useful because it is defined on the positive real line, has a simple functional form, and its mean and variance can be tuned independently of each other. Note that while static extrinsic noise has no impact on the short-time dynamics, it does change the long-time behaviour (compare points obtained using the SSA with static extrinsic noise versus solid lines obtained from solving the rate equations with parameter values equal to the mean of the gamma distributions used for the SSA). Note that here we have assumed that the initial state $j$ at $t = 0$ does not vary from cell to cell. This variability may not be large because transcriptional repressors can sustain a particular inactive state[57]. More general initial gene state distributions are discussed later. Unless otherwise stated, in the rest of the paper, we consider the initial gene state to be the same in all cells (Dirac delta initial gene state distribution).

Thus far we have derived results for short-time behaviour assuming the $N$-state model. This general model can be extended further to account for more complex initiation mechanisms. Thus we consider a model of the type:

$$G_i \underset{k_{j,i}}{\overset{k_{i,j}}{\rightleftharpoons}} G_j, \quad i,j = 1, \ldots, N, \quad G_N \xrightarrow{\rho} G_K + M, \quad M \xrightarrow{d} \emptyset. \quad (5)$$

Here we have $N$ gene states of which only one is active (state $N$). Reversible reactions are allowed between any of pair of states and when transcription occurs the state changes from $N$ to $K$ where the latter can be any integer between 1 and $N$. This state change models reinitiation, where some of the essential transcription factors (but not all) remain bound to the promoter[19]; this can also model promoter proximal-pausing[58]. The $N$-state model (1) is a special case of this model where all gene state reactions are irreversible and no state changes are allowed upon synthesis of a transcript. In Supplementary Note 4 we show using queuing theory that for short times after induction, the model also predicts a mean mRNA count that increases with time as a power law. The exponent is equal to the minimal number of states that are visited between the initial inactive state and the state from which the synthesis of mRNA occurs—hence independent of the connectivity of the reaction pathway describing initiation, the exponent is a lower bound on the number of gene states $N$. In Fig. 3F we show that the theoretical results are in good agreement with the rate equations for the mean mRNA counts in two special cases of reaction scheme (5), one where there is an extra reaction (compared to the $N$-state model) between states 2 and 5 and another one with reversible reactions between states 1 and 2 and a gene state change from 5 to 2 upon transcription.

### The power-law exponent contains useful information on the number of gene states over biologically relevant timescales

To test more directly the practical utility of the theory, we use simulations to mimic the measurement of the mean mRNA count from experimental data. We assumed five measurements of the mean mRNA at regularly spaced time points and that the final measurement was

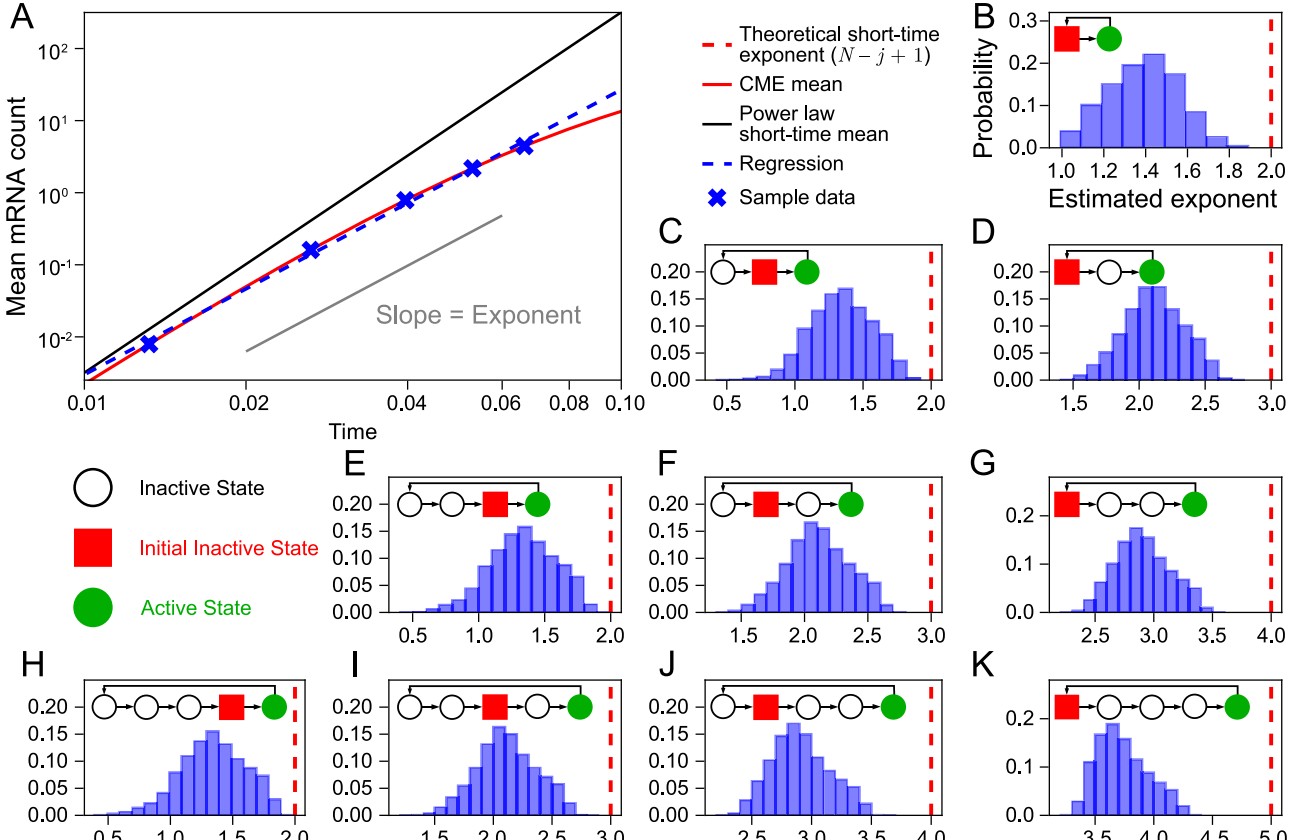

**Fig. 4 | Synthetic data generated using biologically relevant rate parameter values for mammalian cells validates the practical use of the short-time mean mRNA count data to estimate a lower bound for the number of gene states *N*.** Measurements of the mean mRNA count at 5 regularly spaced times were simulated for approximately two million parameter sets through direct numerical integration of the moment equations of the chemical master equation (CME) of the *N*-state model. These time points varied for each parameter set and were chosen such that at the final time point the mean mRNA count per cell was in the biologically realistic range (a few tens). The final time point was also much less than the median half-life of mRNA (several hours). **A** An example curve for one rate parameter set showing deviations of the true power law (dashed blue) from the theoretical power law (solid black) for biologically relevant times. Note that time is non-dimensional since it is multiplied by the mRNA degradation rate: hence *t* = 0.1 corresponds to about one hour for mammalian cells[48]. **B**–**K** For each parameter set, linear regression was used to estimate the power-law exponent from the slope of the log-log plot of the mean versus time. Distributions of the exponent are shown by blue histograms. The model diagram in each figure illustrates the active state (solid green circle), inactive states (open circles) and initial inactive state (solid red square) for each system. Note that *N* (the total number of gene states) varies between 2 and 5 while the initial inactive state *j* varies between 1 and *N* − 1. The exponent *N* − *j* + 1 estimated by our theory (Eq. ([3])) is shown by a vertical red dashed line—in all cases the maximum value of the numerical exponent is below the theoretical value which itself is a lower bound for the number of gene states *N*. See Supplementary Note 2 for simulation details.

taken at a time much earlier than that of the median half-life of mRNA in mammalian cells (several hours). The parameters of the *N*-state model were then chosen such that the mean mRNA at the largest time point was within the biologically meaningful range (less than a few tens of mRNA molecules per cell). For further details on the selection procedure see Supplementary Note 2.

For each of the rate parameter sets, we integrated the time-evolution equations for the mean mRNA counts of the *N*-state model with an initial condition of zero mRNA and starting from inactive state *j* to calculate the mean mRNA at each of the five-time points. Then we used a linear regression on the log-log plot of the mean versus time to estimate the slope and hence exponent of the power law. In all cases, the coefficient of determination of the best linear fit through the data points was larger than 0.99 thus firmly establishing the existence of a power law—however as clear from the example shown in Fig. [4]A, because of the length of the time interval used, the fitted power law was different than the theoretical one and hence it is of interest to understand the relationship between the two.

In Fig. [4]B–K we show histograms of the exponent (computed over all rate parameter sets) for *N*-state models with *N* = 2 − 5 and *j* = 1, . . , *N* − 1. The distributions of exponent values are wide but the maximum value of the exponents is always less than the theoretical exponent

*N* − *j* + 1 (shown by a vertical red dashed line). This is because the next-to-leading order term in the series expansion of the mean in powers of time has a negative sign (see 'Methods') and hence the leading-order term proportional to *t*^{*N*−*j*+1} always overestimates the true mean mRNA count. Hence we have shown using biologically realistic rate parameter constraints that independent of *N*, *j* and the time interval over which the data is collected, the slope calculated from log-log plots of the mean versus time is a reliable means to estimate the lower bound of the number of gene states *N*.

However, it is important to remember that these results rest on four main assumptions: (i) the number of cells is infinite—this is implicit in the calculation of the mean mRNA count from the CME; (ii) linear regression on the log-log plots is the optimal method to infer the exponent; (iii) all mRNA molecules in each cell are detected; (iv) the initial distribution of gene states is a Dirac delta function centred on one of the inactive states.

To address the limitations (i) and (ii) we used the following simulation protocol. We first chose a subset of 20 different parameter sets from those previously analyzed in Fig. [4]. These are distinguished from each other by the mean mRNA count at the last time point which varies between 1 and 25 to capture the realistic range of mRNA abundance in mammalian cells (the median mRNA count per cell for a population of

exponentially growing mouse fibroblasts is approximately 17[48]). For each parameter set, we used the SSA to generate a dataset of mean versus time (sampled at 5-time points) for a finite population of $M$ cells and linear regression was used to obtain the exponent from the log-log plots, as before. Repeating this procedure 5000 times leads to a distribution of exponent values. In Supplementary Fig. 3 we summarise these distributions by means of boxplots for the $N$-state model with $N = 5, j = 4$ and for populations of sizes $M = 100, 500, 1000$ and $10,000$. The medians of all boxplots are in good agreement with the exponent estimated assuming an infinite population size (Fig. 4). Except for populations of 100 cells, the statistical dispersion as measured by the interquartile range is small indicating that for sample sizes of 500 or more cells, uncertainty in the exponent estimates due to finite sample size effects are negligible. This conclusion is valid independent of the mean mRNA count at the last time point, i.e. the method is robust even if the mean count is of order 1. Similar results are obtained when the inference is repeated using non-linear regression applied to the mean count versus time plots (Supplementary Fig. 4) and for the $N$-state model with $N = 5, j = 1$ (Supplementary Figs. 5–6).

To address limitation (iii) we repeated the analysis for the case of non-perfect capture of mRNAs by single-cell mRNA measurements using smFISH or scRNA-seq. Given an integer mRNA count $n$ for a cell at a time point from the SSA, we generated the corresponding observed mRNA count by drawing a binomial random number with success probability $p$ and number of trials equal to $n$[59]. Note that the success probability is a measure of detection efficiency — it is over 80% for FISH experiments[60] and about 5–20% for single-cell sequencing experiments[61–63]. In Supplementary Figs. 7–10 we show that for a typical sample size of 1000 cells, the estimated exponent obtained by linear or non-linear regression from simulated data for the $N$-state model with $N = 5, j = 1$ and $N = 5, j = 4$ is practically independent of the success probability $p$ and is similar to that estimated for an infinite number of cells with perfect detection efficiency in Fig. 4.

To address limitation (iv) we consider cell-to-cell variability in the initial gene state (non-Dirac-delta initial state distributions). The theory shows that the exponent of the power law describing the short-time dynamics sensitively depends on the initial gene state distribution, specifically on which inactive gene states are inaccessible pre-induction (see Supplementary Note 3). However, by simulating the post-induction measurement of mRNA count data over realistic time regimes and sample sizes, we show that independent of the width and type of the initial gene state distribution, the exponent estimated from linear or non-linear regression is still bounded from above by the total number of gene states $N$ (Supplementary Note 5 and Supplementary Figs. 11–12).

In summary, we make two crucial assumptions of our idealised experimental setup, i.e. the time delay for the stimulus to reach the nucleus is known or can be separately measured, and the mean mRNA prior to induction is very small. If additionally, the sample size is at least several hundred of cells, the inferred exponent (and hence a lower bound on the number of gene states) is accurately inferred by means of linear or non-linear regression. This conclusion holds even if the mean expression levels at the last time point are very low or if the measurement method has low detection efficiency.

## Confirmation of the power law in eukaryotic data and estimation of the number of rate-limiting steps for initiation, splicing, and export

To verify that the predicted power law is indeed observed in experimental data, we considered the data set from ref. 64. This data set consists of simultaneous measurements of nuclear and cytoplasmic mRNA following the induction of two genes, *CTT1* and *STL1*, involved in the osmotic stress response of *Saccharomyces cerevisiae*[64]—for an illustration of the main processes mediating this response see Fig. 5A. The experiment used 0.2mol and 0.4mol NaCl to induce this response. To quantify the number of mRNA molecules for each gene, in the

nucleus and the cytoplasm of each imaged cell, smFISH was used at several time points over 60 minutes. This led to a total of eight data sets (2 stress conditions × 2 genes × 2 RNA species). Each temporal measurement was calculated using over a thousand cells (see Supplementary Table 6) thus this dataset is large enough for accurate gene state inference using our method.

However, in this case, we cannot directly fit the power law given by Eq. (3) to the mean count versus time data because (i) the time delay from osmotic shock to DNA binding of the transcription factor(s) mediating the response is unknown and (ii) the mean mRNA is not exactly zero initially. In Supplementary Note 4 we extend the model to take into account these features. The equation for the mean mRNA count for short times is now modified to

$$\langle m(t) \rangle - \langle m(0) \rangle e^{-dt} = \begin{cases} 0 & t < t_0, \\ A(t - t_0)^n + O(t^{n+1}) & t \geq t_0, \end{cases} \quad (6)$$

where $t_0$ is a deterministic time delay, $\langle m(0) \rangle$ is the mean mRNA at time 0 (when the stimulus is first applied), $d$ is the degradation rate of the mRNA and $A$ is some constant. As before, the exponent $n$ equals the number of states visited from the initial inactive state to transcript production. Eq. (6) cannot be reduced to the form of a straight line in log-log space and hence in this case we can only estimate the exponent using non-linear regression. Hence in principle, we need to simultaneously estimate 5 parameters: $d$, $t_0$, $n$, $A$ and $\langle m(0) \rangle$. Note that we cannot simply use the initial mean measured in experiments as the value for $\langle m(0) \rangle$ because there is uncertainty in this value due to the finite number of cells. However, we can safely assume $d = 0$ and only infer 4 parameters which enhances the accuracy of the rest of estimated parameters. The reasons for this are as follows. From Eq. (6) we see that the effect of degradation manifests most clearly on the mean mRNA count for $t < t_0$ because for later times the power law will dominate. However, inspection of the mean mRNA curves versus time for nuclear (Fig. 5C–F) and cytoplasmic data (Fig. 5K–N) shows that the delay time is roughly $t_0 \approx 2$–$4$ min and over this time there is no appreciable degradation of the initial mRNA. This is because the initial mean mRNA count is very small and also because the median mRNA lifetime in yeast is about 11 min[65] which is considerably longer than $t_0$.

Hence the final estimation procedure involves (i) fitting Eq. (6) with $d = 0$ to the experimental mean mRNA curves using non-linear regression (for an illustration see Fig. 5B); (ii) repeating the procedure using replicas of the single-cell data generated using bootstrapping to estimate uncertainty in the parameters. Note that we do not use all data points for the fitting; the optimal number of points is selected by an automated procedure based on the residual sum of squares (Supplementary Fig. 13).

The best-fit models resulting from step (i) of the fitting procedure are shown by the coloured curves in Fig. 5C–F for nuclear mRNA and in Fig. 5K–N for cytoplasmic mRNA for the two genes in two different stress conditions. The sampling distributions of the exponent $n$ resulting from step (ii) of the fitting procedure are shown in Fig. 5G–J for nuclear mRNA and in Fig. 5O–R for cytoplasmic mRNA. The sampling distributions for the rest of the parameters are shown in Supplementary Fig. 14. In all cases, the experimental data is well fit by the model given by Eq. (6) with $d = 0$ thus verifying that mRNA kinetics follows a power law for a short time after transcription initiation begins in the nucleus.

Our results show that the incorporation of an explicit time delay is important and that neglecting it leads to a larger estimate of the exponent and hence an inflated estimate of the number of gene states. For example in Supplementary Fig. 15 we show that if we do not account for the time delay then the exponents for Fig. 5C, K are estimated to be 3.66 and 4.39 respectively (instead of 1.92 and 2.90, respectively).

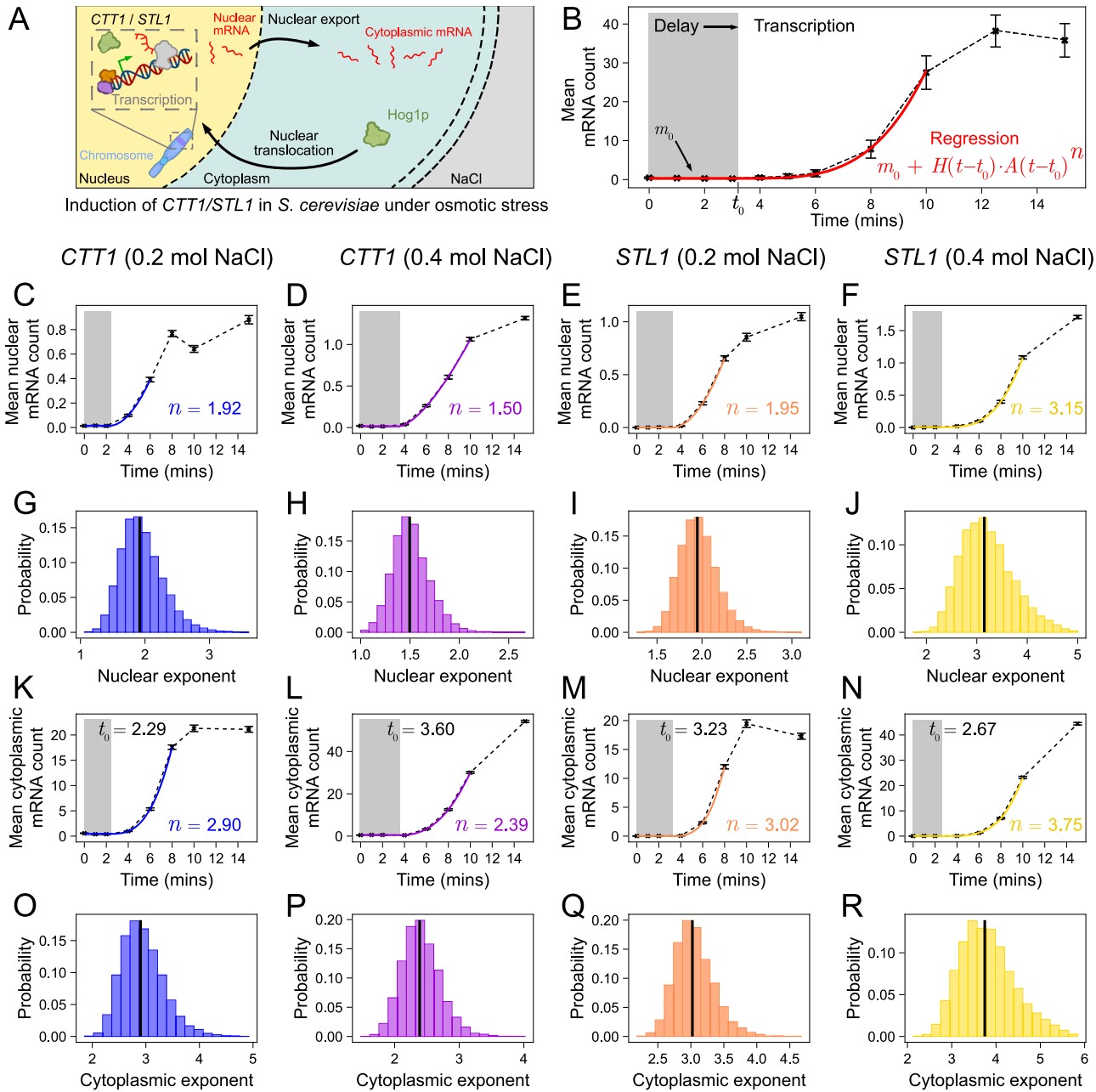

**Fig. 5 | Non-linear regression fitting of mRNA induction curves confirms the short-time power law behaviour in eukaryotic cells. A** Schematic showing the induction of *CTT1* and *STL1* genes in *Saccharomyces cerevisiae* under osmotic stress[64]. This involves the translocation of a kinase (Hog1p) from the cytoplasm to the nucleus where it interacts with transcription factors and chromatin modifiers. **B** Fit of the model given by Eq. ([6]) with $d = 0$ and $m_0 = \langle m(0) \rangle$ using non-linear regression to a typical dataset allows the inference of the exponent $n$ and the rest of the parameters. **C–F** Mean nuclear mRNA count induction curves (crosses joined by black dashed lines) and the best fit model using regression (colour) with the exponent shown next to it. Grey region shows the delay period prior to initiation.

**G–J** Sampling distributions of nuclear mRNA exponents estimated using bootstrapping. **K–N** Mean cytoplasmic mRNA count induction curves (crosses joined by black dashed lines) and the best-fit model using regression (colour) with the exponent shown next to it. **O–R** Sampling distributions of cytoplasmic mRNA exponents estimated using bootstrapping. The median of each distribution computed from 10,000 independent bootstrapped samples is shown by a vertical black line. All error bars show the mean mRNA count ± SEM. Note that (**C**, **G**, **K** and **O**) are for *CTT1* (0.2 mol NaCl); (**D**, **H**, **L** and **P**) are for *CTT1* (0.4 mol NaCl); (**E**, **I**, **M** and **Q**) are for *STL1* (0.2 mol NaCl); (**F**, **J**, **N** and **R**) are for *STL1* (0.4 mol NaCl). See Supplementary Note 2 for technical details.

The median nuclear exponents for *CTT1* and *STL1* in 0.2 mol NaCl were 1.92 and 1.95. From the results of the exponent estimation on the synthetic data, we know that a median exponent of about 2 is consistent with either 3 (Fig. [4]D) or more gene states (Fig. [4]F, I). The median nuclear exponents for *CTT1* and *STL1* in 0.4 mol NaCl were 1.50 and 3.15. Again using the results of simulated data we see that *CTT1* is consistent with at least 2 gene states (Fig. [4]B, C, E, H) while *STL1* is consistent with at least 4 states (Fig. [4]G, J). Hence the simplest model

that can explain all the four nuclear datasets is one with four states. We also note that the difference between the median cytoplasmic and nuclear exponents is approximately one implying that nuclear export kinetics is a first-order process (compare Fig. [3]A and D, and also see 'Methods').

In ref. [14] a multi-state stochastic model with multiple active gene states (connected by reversible reactions) was successfully fitted to data very similar to that of the yeast data used in our analysis, except

that the mRNA was not separated into nuclear and cytoplasmic components. Using maximum likelihood estimation with cross-validation, they found that 2-state and 3-state models were too simple, while a 5-state model structure was prone to over-fitting; hence they settled on an optimal model with four states. In a later study[37] the model (with four states) was refined by performing inference using the measured joint distribution of nuclear and cytoplasmic mRNA counts – the values for the inferred rate parameters can be found in Tables S3–S4 in ref. 37. Inspection of the inferred transcription rates from each gene state for both *CTT1* and *STL1* shows that the rate from one gene state is approximately 5–1000 times larger than the rate from the three other gene states. Hence the 4-state model with four active states is well approximated by a simpler model in which there are three inactive states and one active state which is the 4-state model suggested by the fitting of the power law model above.

We have also analyzed a second dataset consisting of simultaneous measurements of (unspliced) pre-mRNA and (spliced) mRNA following tumour necrosis factor (TNF) induced expression of several inflammatory genes in mouse fibroblasts and bone marrow-derived macrophages (BMDM)[66]. Using quantitative PCR (qPCR) to detect a distinction between pre-mRNA and mRNA, the induction curves for several inflammatory genes were measured (see Fig. 2 and Fig. S2 of ref. 66). It was more challenging to fit Eq. (6) to this data for the following reasons. For three of the induction curves, the estimation gave us the non-physical result of zero delay between stimulus and the start of transcription initiation in the nucleus and a correspondingly high value of the power law exponent. This issue stems from the low number of measurements in the period of time between induction and the setting in of non-power law dynamics. In Supplementary Fig. 16 we show three examples where the fitting was successful. Comparing Supplementary Fig. 16A–B (pre-mRNA and mRNA of gene *CXCL1*) we find the difference in exponents to be 0.82. By the theoretical results in 'Methods', since the difference in exponents is close to 1 (not zero) it follows that splicing, like nuclear export, is consistent with a first-order process and not one with deterministic delay. Supplementary Fig. 16C shows that the exponent from fitting the mRNA data of *ICAM-1* is $2.75 \pm 0.36$; this suggests that the exponent for unspliced mRNA is approximately 2 which, by the results of Fig. 4D, F, I, implies 3 or more gene states.

## Discussion

In this paper we have shown that (i) in most cases, steady-state mRNA count data is not sufficient to estimate the number of inactive states in transcription; (ii) when the steady-state mRNA distribution of the telegraph model distribution well fits the distribution of the *N*-state model ($N > 2$), there exists a simple relationship between the rate parameters of the two models; (iii) for a short time after induction, the increase in the mean mRNA counts follows a universal power law whose exponent provides a lower bound on the number of gene states−this is valid for a large family of models of gene expression, and it is independent of the values of the rate parameters of the gene expression model and the size of static extrinsic noise. We confirm this power law by analysing published yeast and mammalian data. In agreement with estimation using other methods[14,20], our results suggest that the number of inactive eukaryotic gene states can be larger than one.

We note that while (i) was previously suspected, there was no conclusive evidence for it. The success of the telegraph model to fit mRNA count distribution measured for a large number of genes[11–13,24–26] could be due to two distinct reasons. Either the number of rate-limiting steps for most genes is very small, i.e. one for gene inactivation and one for gene activation, or else the number of rate-limiting steps is larger but the telegraph model is the simplest of the *N*-state models that can describe the shape of the steady-state distribution of mRNA counts that are commonly measured. In support of the latter reasoning, ref. 38

showed that the maximum Hellinger distance (HD) computed between the steady-state count distribution of the 3-state model and an effective telegraph model (which spent the same mean time in the active and inactive states as the 3-state model) was quite small. In further support of this, in ref. 67 a mapping was developed between the rate parameters of this model and those of more complex models with the same number of gene states but involving feedback loops which led to good agreement between the time-dependent distributions of both models. As well, more recently it was found that the telegraph model well fits the steady-state distributions of the same model but with static extrinsic noise[40,68]. Our results add further evidence suggesting a significant difficulty in distinguishing between the telegraph model and more complex models. We found that for 3 of the 4 possible types of distribution shapes of the *N*-state models with $N > 2$, i.e. unimodal with Fano factor <2 (shape I), unimodal with Fano factor ≥2 (shape II) and bimodal distributions with zero and non-zero peaks (shape III), the telegraph model typically provides an excellent fit. The underlying theoretical reasons for this are that for shapes I-III, the distributions of the *N*-state model look like simple distributions which are limiting cases of the telegraph model, e.g. the Poisson distribution with Fano factor = 1 (zero inactivation rate), the negative binomial distribution with Fano factor >1 (the inactivation rate is much larger than the activation rate) and the sum of a Dirac delta function and a Poisson distribution with a non-zero mode (small activation and inactivation rates, combined with large enough transcription rate). While these distributions may explain the agreement between the *N*-state and the telegraph model in some cases, generally this explanation is not sufficient because we have shown that within each shape type there is quite a variation in distribution type (Supplementary Fig. 1). What is perhaps more intriguing is that we found the existence of a distribution shape of the *N*-state models with $N > 2$ (shape IV) which to our knowledge has not been described before and which cannot be fit by any rate parameter choice of the telegraph model−this distribution has two peaks, both of them at a non-zero value. We note that an extension of the telegraph model to include transcription from both gene states (the leaky telegraph model[69]) can predict a qualitatively similar distribution shape (in the limit of slow switching between the two states). This implies that fitting such a model to steady-state count data characterised by distributions of shape IV may erroneously lead one to infer two active states when in reality there is only one active state.

We also established a useful relationship between the rate parameters of the 2-state and *N*-state models with $N > 2$ when the steady-state mRNA count distributions of the two models are similar. In particular Eq. (2) provides an interesting insight: the inverse of the activation rate of the telegraph model underestimates the total time spent in the inactive states $1, \ldots, N - 1$ of the *N*-state model, i.e. $\langle \tau_{1 \to N} \rangle$. Similarly, the inverse of the inactivation rate of the telegraph model underestimates the time spent in the active state *N* of the *N*-state model, i.e. $1/k_N$. The underestimation is due to the factor $2/(\mathrm{CV}_{1 \to N}^2 + 1)$ which is bounded between 1 and 2 because $\mathrm{CV}_{1 \to N}^2 \leq 1$ and $\lim_{N \to \infty} \mathrm{CV}_{1 \to N} = 0$. This also implies that the relative error in the estimates of the time spent in the inactive and active states is at most 50%. We note that the factor $2/(\mathrm{CV}_{1 \to N}^2 + 1)$ naturally emerges from requiring the variance of the interarrival time between successive mRNA synthesis events of the 2 and *N*-state models to be matched, which as argued earlier is a necessary condition to ensure matching of the distribution of mRNA counts. Our results hence suggest that fitting the telegraph model to steady-state mRNA count data from a gene with a larger number of states leads to a systematic bias in the estimated speed of promoter switching. This is interesting considering that static extrinsic noise has also recently been shown to lead to systematic over- or underestimation of rate parameters inferred from steady-state count data[40]. Overall these studies imply that generally reliable rate parameter estimation from steady-state count data is difficult unless extrinsic noise sources are identified and properly taken into account.

Clearly, the best way to overcome the limitations of steady-state data is to use temporal data. Previously, a few methods have utilised the information contained in the distributions of mRNA counts measured at various time points[14,15,37] to simultaneously estimate the rate parameters and the number of gene states. Motivated by the fact that this data is not commonly available and the inference methods used are rather intricate, we here developed a much simpler method which utilises only the mean mRNA count measured for a short time after gene induction. With regard to whether dynamical low-order moment data is sufficient to distinguish between gene expression models with varying numbers of states, we are not aware of any study which has studied this question before; an exhaustive study of moment-based and distribution-based inference approaches compared their accuracy to estimate rate parameters at a fixed number of gene states[37]. The main advantages of our method are the ease with which the input data (the mean mRNA counts) can be measured (qPCR, smFISH, single-cell and bulk RNA sequencing) and with which the number of gene states is estimated. In particular the latter follows directly from the exponent of the power law inferred by regression analysis on mean mRNA levels over time. The method can only estimate a lower bound for the number of gene states, not the true number of states—this limitation is common to all inference methods since it is only possible to identify the simplest $N$-state model that is consistent with the data. The accuracy of the inferred lower bound is high provided: (i) the time delay between the application of the stimulus and the start of transcription initiation can be well estimated; (ii) the number of cells is at least several hundred; (iii) a sufficiently large number of measurements is taken in the short period of time following gene induction. If these conditions are met then the method retains its accuracy even if the mean expression levels at the largest time point are very low (naturally or because of low detection efficiency) or if there is considerable static extrinsic noise. Of the three aforementioned limitations, the most challenging is the first—we found the easiest way to overcome this is to incorporate an explicit description of time delay in the model and then to use non-linear regression to estimate this quantity simultaneously with the lower bound for gene states. The exponents from nuclear and cytoplasmic mean count data suggest four gene states (three inactive states) for two yeast genes (in agreement with inference by a far more complex inference method[14]), and one rate-limiting step for nuclear export (in agreement with observations using single molecule microscopy[70,71]). Furthermore, the exponent from spliced and unspliced mRNA count data for a mammalian gene suggests one rate-limiting step for splicing. We found no evidence that nuclear export or splicing can be described by an effective reaction step with a fixed time delay. These results are broadly in agreement with a recent model-based analysis of single-cell and single-nucleus RNA sequencing data[72]. Importantly, our analysis shows that failing to account for post-transcriptional processing and the delay between stimulus and initiation will lead to an overestimation of the number of gene states.

Finally, one may ask what is the practical utility of determining a lower bound on the number of gene states using the method that we have here developed. Transcriptional regulation involves several distinct steps, and it remains unclear which of these are targets of regulation and how these steps have emerged. As noted earlier, the number of inactive states can be seen as being equal to the number of rate-limiting reaction steps in initiation, which are likely the key control points of transcriptional regulation. The method described allows one to rapidly estimate a lower bound for the number of these points. If the experiment is repeated with different perturbations then one can possibly establish a biological interpretation for each of the discrete gene states. So for example, say we wanted to understand the rate-limiting steps in mammalian transcription. In vitro experiments have identified six transcription factors (TFIID,

TFIIA, TFIIB, TFIIF, TFIIE and TFIIH) that must bind in a certain order before the preinitiation complex begins elongation[1]. The fact that the number of essential transcription factors is significantly larger than the number of states of models commonly fit to mammalian gene expression data[20,25] suggests that only the binding of a few transcription factors is rate-limiting. We can then setup a series of induction experiments which differ from each other only by the overexpression of one transcription factor. In all of these experiments, we assume the same inducer is used such that the distribution of the initial gene state remains narrow and centred on the same value. If a transcription factor is under normal conditions associated with a rate-limiting step, when it is over-expressed it will not be anymore since the reaction will be sped up. In this case, we expect the number of gene states to decrease by one (compared to that in normal conditions) which will result in a similar decrease in the exponent of the power law estimated from the short-time kinetics.

Concluding, we have shown that important details of transcription such as the number of rate-limiting steps in initiation have signatures in the mean mRNA data collected a short time after gene induction. The method's simplicity, particularly its avoidance of the need for the distribution of mRNA counts and of sophisticated inference techniques, shows that simple mathematical tools can also be used to gain insight from transcriptomic data.

## Methods

### Mapping between the rate parameters of the telegraph model and the $N$-state model

We consider a mapping between the rate parameters of an effective telegraph model and those of the $N$-state model, whose reaction scheme is given by Eq.(1). This mapping involves the waiting time distribution between successive mRNA synthesis events.

**Moments of the waiting time distribution.** Let $\tau$ denote the waiting time between successive mRNA synthesis events in the $N$-state model, and $f_N(\tau)$ its corresponding probability density function. The moments of $f_N(\tau)$ follow directly from the definition of the Laplace transform of $f_N(\tau)$, denoted by $f_N^*(s)$, since

$$f_N^*(s) = \langle e^{-s\tau} \rangle_N = \sum_{n=0}^{\infty} (-1)^n s^n \frac{\langle \tau^n \rangle_N}{n!}. \tag{7}$$

Suppose that the gene is in the active state $N$ at $t = 0$. Let $w_{i,j}(t)$ denote the mixed joint probability density function that the gene spends time $t$ in state $i$ before moving to state $j$. The probability distribution function of the total time to move from state $N$ to 1, and then back to state $N$ through states 2, 3, ..., $N-1$ is the convolution of the individual distributions $w_{N,1}(t), w_{1,2}(t), ..., w_{N-1,N}(t)$. Equivalently, the Laplace transform of the probability density function of the total time is given by

$$w_{N,1}^*(s) w_{1,2}^*(s) \cdot \cdots \cdot w_{N-1,N}^*(s), \tag{8}$$

where $w_{i,j}^*(s)$ denote the Laplace transform of $w_{i,j}(t)$. Since the mRNA is transcribed only with some probability once the state $N$ is reached, it follows that any number $n \geq 0$ of the cycles ($N$ to 1 to 2 ...to $N$) is permitted before the mRNA is synthesised. This implies that the Laplace transform of the waiting time distribution between two successive mRNA synthesis events is given by

$$f_N^*(s) = \sum_{n=0}^{\infty} [w_{N,1}^*(s) w_{1,2}^*(s) \cdot \cdots \cdot w_{N-1,N}^*(s)]^n w_{N,N}^*(s)$$

$$= \frac{w_{N,N}^*(s)}{1 - w_{N,1}^*(s) w_{1,2}^*(s) \cdot \cdots \cdot w_{N-1,N}^*(s)}, \tag{9}$$

where

$$w^*_{i,i+1}(s) = \frac{k_i}{s+k_i}, \quad i=1,\ldots,N-1,$$
$$w^*_{N,1}(s) = \frac{k_N}{s+k_N+\rho}, \quad w^*_{N,N}(s) = \frac{\rho}{s+k_N+\rho}. \tag{10}$$

Note that $w^*_{N,N}(s)$ is the Laplace transform of the distribution of the time spent in state $N$ before an mRNA transcription event occurs. We also note that the series in Eq. (9) is guaranteed to converge since $w^*_{N,1}(s)w^*_{1,2}(s)\cdot\cdots\cdot w^*_{N-1,N}(s) < 1$ for non-negative $s$. Substituting Eqs. (10) into (9) gives

$$f^*_N(s) = \frac{\rho\prod_{i=1}^{N-1}(s+k_i)}{(s+k_N+\rho)\prod_{i=1}^{N-1}(s+k_i) - \prod_{i=1}^{N}k_i}. \tag{11}$$

This expression can be inverted to get $f_N(t)$ using partial fraction decomposition. The value of $f_N(0)$, however, can be obtained directly from $f^*_N(s)$ using the initial value theorem, yielding

$$f_N(0) = \lim_{s\to\infty}[sf^*_N(s)] = \rho. \tag{12}$$

From the expression for $f^*_N(s)$ and using Eq. (7), we get for the first two (raw) moments,

$$\langle\tau\rangle_N = -\frac{df^*_N}{ds}\bigg|_{s=0} = \frac{k_N}{\rho}S_{N,1}, \tag{13}$$

$$\langle\tau^2\rangle_N = \frac{d^2f^*_N}{ds^2}\bigg|_{s=0} = 2\left[\left(1+\frac{k_N}{\rho}\right)S_{N,1} - \frac{1}{k_N}\right]\frac{k_N}{\rho}S_{N,1} - \frac{k_N}{\rho}(S^2_{N,1} - S_{N,2}), \tag{14}$$

where

$$S_{N,1} = \sum_{j=1}^{N}\frac{1}{k_j}, \quad S_{N,2} = \sum_{j=1}^{N}\frac{1}{k_j^2}. \tag{15}$$

From here we find that the variance $\sigma_\tau^2 = \langle\tau^2\rangle_N - \langle\tau\rangle_N^2$ is given by

$$\sigma_{\tau,N}^2 = \langle\tau^2\rangle_N - \langle\tau\rangle_N^2 = \frac{k_N}{\rho}S_{N,2} + \frac{k_N}{\rho}\left(1+\frac{k_N}{\rho}\right)S_{N,1}^2 - \frac{2}{\rho}S_{N,1}. \tag{16}$$

We note that the inverse of $\langle\tau\rangle_N$, which is the effective transcription rate, can be interpreted as the actual transcription rate $\rho$ multiplied by the steady-state probability that the gene is in the $N$th state,

$$P_{\text{on},N} = \frac{1}{k_N\sum_{i=1}^{N}\frac{1}{k_i}}. \tag{17}$$

**Rate parameters of an effective telegraph model via moment matching.** We now consider the case of $N=2$ (the telegraph model). Let $x = k_2/k_1$. Then from Eqs. (12), (13) and (16) we have

$$f_2(0) = \rho, \quad \langle\tau\rangle_2 = \frac{1+x}{\rho}, \quad \sigma_{\tau,2}^2 = \frac{2x^2}{k_2\rho} + \frac{(x+1)^2}{\rho^2}. \tag{18}$$

To map the $N$-state model to the effective 2-state telegraph model, we seek $\rho^*$, $k_1^*$ and $k_2^*$ of the telegraph model such that the following waiting time statistics of the two models match:

$$f_2(0) = f_N(0), \quad \langle\tau\rangle_2 = \langle\tau\rangle_N, \quad \sigma_{\tau,2}^2 = \sigma_{\tau,N}^2. \tag{19}$$

From these equations, we get the following expressions for $\rho^*$, $k_1^*$ and $k_2^*$,

$$\rho^* = \rho, \quad k_1^* = \frac{2\left(\langle\tau\rangle_N - \frac{1}{\rho}\right)}{\sigma_{\tau,N}^2 - \langle\tau\rangle_N^2}, \quad k_2^* = \frac{2\rho\left(\langle\tau\rangle_N - \frac{1}{\rho}\right)^2}{\sigma_{\tau,N}^2 - \langle\tau\rangle_N^2}. \tag{20}$$

In this mapping, the effective rate parameters depend only on $\rho$, $k_1,\ldots,k_N$ but not on the degradation rate $d$. It can also be shown that this mapping guarantees that the mean mRNA number is also the same between the models. We also note that both $k_1^*$ and $k_2^*$ are always positive since $\langle\tau\rangle_N > 1/\rho$ and $\sigma_{\tau,N}^2 > \langle\tau\rangle_N^2$, which can be verified from expressions in Eqs. (13) and (16).

The mapping in Eq. (20) can be interpreted by introducing the random time $\tau_{1\to N}$ it takes the gene to go from state 1 to state $N$. Since this time is a sum of independent exponentially distributed random variables with rate parameters $k_1,\ldots,k_{N-1}$, its mean and variance are given by

$$\langle\tau_{1\to N}\rangle = \sum_{j=1}^{N-1}\frac{1}{k_j}, \quad \langle\tau_{1\to N}^2\rangle - \langle\tau_{1\to N}\rangle^2 = \sum_{j=1}^{N-1}\frac{1}{k_j^2}. \tag{21}$$

Substituting these expressions into Eq. (20), we get

$$k_1^* = \frac{1}{\langle\tau_{1\to N}\rangle}\cdot\frac{2}{\text{CV}_{1\to N}^2+1}, \quad k_2^* = k_N\cdot\frac{2}{\text{CV}_{1\to N}^2+1}, \tag{22}$$

where $\text{CV}_{1\to N}$ is the coefficient of variation of $\tau_{1\to N}$,

$$\text{CV}_{1\to N}^2 = \frac{\langle\tau_{1\to N}^2\rangle - \langle\tau_{1\to N}\rangle^2}{\langle\tau_{1\to N}\rangle^2} = \sum_{j=1}^{N-1}\frac{1}{k_j^2}\left(\sum_{j=1}^{N-1}\frac{1}{k_j}\right)^{-2}. \tag{23}$$

**Short-time analysis for the mean mRNA count**
**Derivation of the short-time behaviour of the mean mRNA count following induction.** Assuming Markovian dynamics, the CME for the $N$-state reaction system described by Eq. (1) reads:

$$\frac{d}{dt}P_1(n,t) = d(n+1)P_1(n+1,t) - dnP_1(n,t) + k_NP_N(n,t) - k_1P_1(n,t),$$
$$\frac{d}{dt}P_i(n,t) = d(n+1)P_i(n+1,t) - dnP_i(n,t)$$
$$+ k_{i-1}P_{i-1}(n,t) - k_iP_i(n,t), \quad i\in\{2,N-1\}$$
$$\frac{d}{dt}P_N(n,t) = \rho(P_N(n-1,t) - P_N(n,t)) + d(n+1)P_N(n+1,t)$$
$$- dnP_N(n,t) + k_{N-1}P_{N-1}(n,t) - k_NP_N(n,t), \tag{24}$$

where $P_i(n,t)$ is the probability of observing $n$ mRNA molecules when the gene is in state $i$ at time $t$.

We first derive the moment equations from the CME. For compactness and simplicity of the expressions, we non-dimensionalise time and the reaction rates by the mRNA degradation rate: $t\mapsto td$, $k_i\mapsto k_i/d$ and $\rho\mapsto\rho/d$.

The $r^{th}$ moment of the mRNA, conditional on the gene being in state $i$, is given by:

$$(n^r)_i = \sum_n n^rP_i(n). \tag{25}$$

Note that the case $r=0$ gives the probability of being in gene state $i$. In what follows, unless otherwise stated, for notational convenience we suppress the time dependence of the probabilities and of the moments.

By summing the terms on the right and left-hand sides of Eq. (24) over $n$, we find the time-evolution equations for the probabilities of

being in each gene state:

$$\frac{d}{dt}(n^0)_1 = k_N(n^0)_N - k_1(n^0)_1, \tag{26}$$

$$\frac{d}{dt}(n^0)_i = k_{i-1}(n^0)_{i-1} - k_i(n^0)_i, \quad i \in \{2, N\}. \tag{27}$$

Similarly by multiplying the right and left-hand sides of Eq. (24) by $n$ and summing over $n$, we obtain the time-evolution equations for the first moments of the mRNA counts (conditional on the gene state):

$$\begin{aligned}
\frac{d}{dt}(n)_1 &= -(n)_1 + k_N(n)_N - k_1(n)_1, \\
\frac{d}{dt}(n)_i &= -(n)_i + k_{i-1}(n)_{i-1} - k_i(n)_i, \quad i \in \{2, N-1\}, \\
\frac{d}{dt}(n)_N &= \rho(n^0)_N - (n)_N + k_{N-1}(n)_{N-1} - k_N(n)_N.
\end{aligned} \tag{28}$$

Say the gene is initially in inactive state $j$ where $j \in 1, \ldots, N-1$ and that the number of mRNA is zero. Due to the uni-directional nature of the gene state transitioning processes, the only reaction that can occur for very short time times is $G_j \rightarrow G_{j+1}$. Since at $t = 0$, $(n^0)_j = 1$ and $(n^0)_i = 0$, $\forall i \neq j$, it immediately follows from Eq. (27) that for short times $(n^0)_{j+1} \approx k_j t$.

The time-evolution for the next state to be visited, i.e. $j+2$, is given by

$$\frac{d}{dt}(n^0)_{j+2} = k_{j+1}(n^0)_{j+1} - k_{j+2}(n^0)_{j+2}. \tag{29}$$

Assuming a Taylor series expansion of $(n^0)_{j+2}$ in powers of $t$, integrating the above differential equation and using the previous result for $(n^0)_{j+1}$ it follows that to leading order, $(n^0)_{j+2} \approx \frac{1}{2}k_j k_{j+1}t^2$.

By iterating this procedure, one finds that generally

$$(n^0)_{j+r} \approx \frac{\prod_{i=j}^{j+r-1} k_i}{r!}t^r, \quad r = 1, \ldots, N-j. \tag{30}$$

Now summing Eq. (28), we obtain a time-evolution equation for the total mRNA count:

$$\frac{d}{dt}\langle m \rangle = \rho(n^0)_N - \langle m \rangle, \tag{31}$$

where $\langle m \rangle = \sum_{i=1}^{n}(n)_i$. Assuming a Taylor series expansion of $\langle m \rangle$ in powers of $t$, integrating the above differential equation and using Eq. (30), we finally obtain:

$$\langle m \rangle \approx \rho \frac{\prod_{i=j}^{N-1} k_i}{(N-j+1)!}t^{N-j+1}. \tag{32}$$

Next we derive an estimate of how small the time should be such that Eq. (32) holds. We will do this by deriving the next-to-leading order correction to the mean and then comparing the size of this term with the leading-order term previously derived.

From Eq. (27) it follows that

$$\frac{d}{dt}(n^0)_{j+r} = k_{j-1+r}(n^0)_{j-1+r} - k_{j+r}(n^0)_{j+r}, \quad j+r \in \{2, N\}. \tag{33}$$

Substituting $(n^0)_{j+r} = \alpha_{j,r}t^r + \beta_{j,r}t^{r+1} + O(t^{r+2})$ and collecting terms of order $t^r$, we obtain:

$$\beta_{j,r+1} = \frac{1}{r+2}(k_{j+r}\beta_{j,r} - k_{j+r+1}\alpha_{j,r+1}). \tag{34}$$

Note that $\alpha_{j,r}$ is known; by Eq. (30), it immediately follows that it is given by

$$\alpha_{j,r} = \frac{\prod_{i=j}^{j+r-1} k_i}{r!}. \tag{35}$$

Eq. (34) is a first-order non-homogeneous recurrence relation with variable coefficients. This can be solved using standard methods yielding:

$$\beta_{j,r} = \left[\prod_{i=0}^{r-1}f_i\right]\left(\beta_{j,0} + \sum_{m=0}^{r-1}\frac{g_m}{\prod_{i=0}^{m}f_i}\right), \tag{36}$$

where

$$f_r = \frac{k_{j+r}}{r+2}, g_r = \frac{-k_{j+r+1}\alpha_{j,r+1}}{r+2}. \tag{37}$$

Furthermore from Eqs. (26) and (27) and the initial conditions $(n^0)_j = 1$, and $(n^0)_i = 0$, $\forall i \neq j$, it follows that

$$(n^0)_j = 1 - k_j t + O(t^2),$$

which implies that

$$\beta_{j,0} = -k_j. \tag{38}$$

Hence from Eqs. (35), (36), (37) and (38) it follows that

$$\begin{aligned}
(n^0)_N &= \alpha_{j,N-j}t^{N-j} + \beta_{j,N-j}t^{N-j+1} + O(t^{N-j+2}), \\
&= \frac{\prod_{i=j}^{N-1}k_i}{(N-j)!}t^{N-j} - \frac{\prod_{i=j}^{N-1}k_i}{(N-j+1)!}\left(\sum_{i=j}^{N}k_i\right)t^{N-j+1} + O(t^{N-j+2}).
\end{aligned} \tag{39}$$

Finally substituting this result in the time-evolution for the total mRNA, Eq. (31), and computing the series solution we obtain:

$$\langle m \rangle = \rho\left[\prod_{i=j}^{N-1}k_i\right]\left[\frac{t^{N-j+1}}{(N-j+1)!} - \left(1 + \sum_{i=j}^{N}k_i\right)\frac{t^{N-j+2}}{(N-j+2)!}\right] + O(t^{N-j+3}). \tag{40}$$

Comparing the magnitudes of the terms proportional to $t^{N-j+1}$ and $t^{N-j+2}$, we find that the leading-order term, i.e. Eq. (32), is accurate provided

$$t \ll \frac{N-j+2}{1 + \sum_{i=j}^{N}k_i} \le N+1, \quad j \in \{1, N-1\}. \tag{41}$$

The result suggests that the leading-order power law is visible over a long period of time provided the number of gene states to be traversed from the the initial inactive state to the active state $(N-j)$ is not too small and provided the rates of hopping from one gene state to another (relative to the mRNA degradation rate) are not too large. Note that Eq. (41) does not involve $\rho$ which implies that increasing $\rho$ increases the max mRNA count reached at the end of the time interval over which the leading-order power law is valid, without affecting the length of this interval.

**Post-transcriptional processing of mRNA.** We note that in the previous calculations, we have interpreted the mRNA ($M$) in the $N$-state model as being the total mRNA. In some experiments, it is possible to distinguish between spliced and unspliced mRNA[66] and between nuclear and cytoplasmic mRNA[64]. We can model post-processing by

adding extra reactions to the scheme described by Eq. (1). For example, if we interpret $M$ as being unspliced nuclear mRNA then we could add a first-order reaction $M \to M_s$ describing the synthesis of spliced nuclear mRNA $M_s$ with some rate $k_s$. Since for short times, $d/dt\langle m_s \rangle \propto k_s \langle m \rangle$, it immediately follows by integrating Eq. (32) that the mean spliced mRNA count will follow a power law with an exponent that is one larger than that for unspliced mRNA. By the same argument, the mean cytoplasmic mRNA will follow a power law with exponent that is one more than that of nuclear mRNA. If we interpret the mRNA in the $N$-state model as the cytoplasmic mRNA, then one could add a processing step to describe its translation to protein; in that case, the short-time power law exponent for the latter will be one more than that for the cytoplasmic mRNA. On the other hand, if the post-processing of $M$ to a species $M_s$ can be modelled by a deterministic delay reaction (with delay $T$) then $\langle m_s \rangle(t) = \langle m \rangle(t - T)$ and hence in this case the power law exponent is unaffected by processing.

### Reporting summary
Further information on research design is available in the Nature Portfolio Reporting Summary linked to this article.

### Data availability
The synthetic data generated in this study have been deposited in Zenodo under accession code [14245774], available at ref. 73. The experimental yeast data used in this study are available at BioStudies with the dataset identifier [S-BIAD1] and are also available at ref. 74.

### Code availability
Code for generating various figures and fitting models to data is available at ref. 73 and on GitHub [https://github.com/agnflame/PowerLaw].

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

## Acknowledgements

This work was supported by a BBSRC EASTBIO PhD studentship to A.G.N. (BB/T00875X/1) and a Leverhulme Trust research award to R.G. (RPG-2020-327). We thank Prof. Peter Swain for his advice and comments in pursuing models of gene induction.

## Author contributions

R.G. conceived the original idea. A.G.N. performed the simulations, fitted the models to the data and produced the figures. A.G.N., J.S.-N., M.R.E. and R.G. derived the mapping between the rate parameters of the telegraph and N-state model. J.S.-N., M.R.E. and R.G. derived the theory describing the short-time dynamics. R.G. wrote the initial draft of the manuscript. A.G.N., J.S.-N., M.R.E. and R.G. contributed to review and editing of the later versions of the manuscript. M.R.E. and R.G. provided supervision of the project.

## Competing interests

The authors declare no competing interests.
