## [Transparent Peer Review file · Nature Communications]

Transient power-law behaviour following induction distinguishes between competing models of stochastic gene expression

Corresponding Author: Professor Ramon Grima

Version 0:

Reviewer comments:

Reviewer #1

(Remarks to the Author)

In the manuscript the authors study the problem of the identifiability of the structure of transcriptional states from mRNA count data. On one hand the authors show that steady state distributions of mRNA a classic telegraph model is virtually indistinguishable from more sophisticated multistate models. While this result is interesting, it is not surprising. On the other hand, I found very interesting the second part of the paper, where they show elegantly how time course data of average mRNA number of a gene being induced can be used to extract information about the underlying number of states. Overall the paper is interesting, relevant, well written, and thoroughly documented.

Some more detailed comments/suggestions:

- 1) The authors made clear (using relevant experimental references) that parameters used along the manuscript are based on relevant ranges. Nevertheless, it is not clear how some of the cases were chosen. For instance the parameters used for Fig.3 A-D. Is there any rationale behind this choice?
- 2) I did not fully understand the scheme of panel F ^t4. Is the scheme a state in which the successful production of mRNA returns the system to state 2 with rate $\nu=1600$, or it returns to the state 1 with rate $k_5=0.5$? If this is the case, aren't the rate to each other, and the result would be the same as not having the 1st state? Rate $k_{21}=0.5$ seems also to be very slow.
- 3) One of the experimental problems with mRNA counting is the detection accuracy. This results in the observations of only a fraction of the mRNAs being counted. How would this affect the results of the theory? As a first approximation I think the theory should hold. Assuming that the average mRNA number measured is a fraction of the real number of mRNA, it would preserve the exponent of the power law. On the other hand, due to many cells with low numbers of mRNA at early time points will bias this result.
- 4) The power law fits (Fig. 4) only allow to predict a lower bound for the number of states from the initial state of the gene. Nevertheless, Fig. 4 seems to confirm that it is not any lower bound, but actually a good predictor of the number of states. For almost all the cases the next integer to the float number of the predicted slope is the correct answer. I think this is a strong result! Do the authors predict that this result will hold in real situations?
- 5) The power law fits for real data in Fig.5 do not use a log-log scale as employed in the rest of the manuscript. I suspect that the authors chose this visualization because the measured number of mRNA is too low (or even zero) at early timepoints, resulting in large deviations from the fit in the log-log plot. How did the fitting took this into account? If the fitting was done in the linear scale, that means that initial points were not used to fit the slope of the power law, so effectively all the information of the fitting is concentrated in the last 3 timepoints. Related to this, how was the error calculated for the exponents in this fitting? I find that clarity in this section is relevant, since it illustrates the applicability of the method.

(Remarks on code availability)

The code is presented in a jupyter notebook using Julia written and commented clearly. The output plots follow the same

style as the ones shown in the paper making very easy to reproduce, use, and recycle the code.

Reviewer #3

(Remarks to the Author)

In the manuscript, "Transient power-law behaviour following induction distinguishes between competing models of stochastic gene expression", authors Nicoll et al describe a simple analysis of the mean expression of gene activity in the initial time following application of an induction signal. The authors argue that if (1) all cells start in a fixed initial state and (2) there is only one fully active transcription state, then (at least for a short time following stimulus) the mean expression level will follow a power-law behavior. Moreover, the exponent of this power-law will provide an easily calculated lower bound on the number of steps between the initial state and the transcriptionally active state.

Overall, this is an interesting paper, and I can certainly see value in having simple means to infer models for gene expression dynamics. However, I have several concerns about the manuscript in its current form as follows:

Major Concerns

The manuscript lacks sufficient discussion of uncertainty quantification in their analyses of experimental data. Without these details and a thorough quantification of posterior parameter uncertainties after fitting, it is difficult to determine if the author's claims that they have identified the number of states is sufficiently well supported.

The statements, "Consider an experimental setup whereby some gene of interest is inactive before induction by an inducer molecule. From application of the inducer to its binding to DNA, there is some time lag due to nuclear translocation and other processes. Hence we shall consider $t = 0$ to be the point at which induction starts in the nucleus." seems to assume that one can precisely know the time at which the inducer reaches the gene of interest. But this can be a complicated process. For example, in the data analyzed in Fig 5A-D, gene expression is induced by a time varying MAPK signal that goes through multiple phosphorylation steps before reaching the nucleus. The timing of this process could be unknown (and is possibly different for every cell), but appears to be quite fast for the case of the yeast osmotic shock. It is not clear from the manuscript (1) how sensitive is the presented analysis to errors in the time $t=0$, or (2) how the presented analysis could be modified to account for this deviation in this time. The authors should analyze how errors in this specific time would affect their determination of the minimum number of states.

I am concerned about the practical aspects of fitting such small values of mean expression. Several plots in Figs 3, 4A, 5 show mean expression levels that are much less than one, and Fig 3 shows corresponding variances of equal magnitude. Unfortunately, this suggests situations where the standard deviations are much greater than the mean. As a result, it might be difficult to determine the mean of the mRNA to any degree of accuracy without measuring a very large number of cells. Moreover, if there is any error in those measurements (e.g., an erroneous spot call in a smFISH image processing step for a small fraction of cells), then those errors could have a major impact on the inferred exponent. Some effort needs to be taken to explore the uncertainty in these values (e.g., add error bars to the curves in Figs 4, 5) and propagate that uncertainty to compute confidence intervals for the identified exponents.

In referring to Fig 4A, the statement, "Note that by the use of these equations, here we made the implicit assumption that the number of cells is infinite (in practice this means the number of cells is larger than a few thousand such that the standard error of the mean is very small)" is not sufficiently clear and requires additional analysis. At least for the representative example depicted in Fig 4A, it would be nicer to propose a reasonable number of cells (e.g., 50, 100, 1000, 10000 per time point), generate the simulated data to find the mean, and then infer their model parameters from that simulated data. At a minimum, the authors could use bootstrapping to then determine the uncertainty of their parameter estimates.

In the application of the approach to the data in Fig 5, there is insufficient discussion of the uncertainty in the data and how that affects parameter estimates in the resulting model. In Fig 5, I would expect to see error bars on all data points. For the data in panels A-D, these could be SEM calculated from the measured single-cell distributions at those times. For the data in E-H, these could include error bars showing the SEM between the different experimental replicas. For Table 1, the description of the approach to quantify the SEM in the parameter estimates does not provide sufficient detail. SI Note 5 only states that the authors used a nonlinear least square fitting routine to fit the measured mean at each time. This description does not include details for how the authors propagate uncertainty in the data during their fitting procedure. The authors need to provide some sort of cross-validation, bootstrapping analysis, or Bayesian posterior quantification to compute confidence intervals or uncertainties in their estimates, given the data.

Considering the issues above, the authors should provide a more detailed discussion of the limitations of their approach. For example, their analysis focusses on a limited case where: (1) the cells start with zero mRNA; (2) cells start in a single fixed state; (3) the process starts at a precisely known initial time; and (4) the mean number of mRNA is measured without measurement noise. It is not clear if/how the presented analysis could be extended to allow for non-trivial initial probability distributions for the states, uncertain initiation times, non-zero initial expression levels, or measurement errors in the quantification of expression. Although it may be unreasonable to tackle all these concerns in a single paper, each of these major assumptions should either be specifically addressed in the manuscript or described at the end as a current limitation that requires future investigation.

Beyond the technical aspects of determining N from data using early time dynamics, it is not clear from the manuscript what is the utility for determining this number of states. Since the concept of discrete states is a convenient simplification of a

much more complicated reality, it would be nice if the manuscript could provide more insight into questions such as: What would one do with the lower bound N once it is determined? How would the knowledge of N this assist the researcher to build more predictive models, design more effective experiments, or discriminate more effectively between competing hypotheses for regulation mechanisms?

Minor Concerns:

I disagree with the statement, "Despite this detailed molecular knowledge, it has thus far proved impossible to quantitatively predict the kinetics of gene expression at the single-cell level." Many of the papers cited have accurately predicted the distributions of single-cell gene expression upon fitting very similar types of models to data.

The sentence in the introduction, "We note that while an extension of this model to include reversible reactions between gene states or transcription from multiple gene states is likely more realistic [17–19], nevertheless it has many more rate parameters than the irreversible model and hence the latter is typically preferred," needs more support for the second clause. Most of the models I have seen that have fit and predicted single-cell data have included reversible reactions, including the ones cited in this specific sentence. Since the authors explore this case later in the manuscript, it would be worth mentioning that this approximation is relaxed later in the paper.

The discussion of the interplay between varying numbers of gene states and the effect of extrinsic noise is not sufficiently clear. The authors describe previous efforts toward finding the optimal number of states, N , needed to capture and predict observed data. In such problems, the goal is to find the model that is best supported by available data (e.g., through analyses using cross-validation or Bayesian estimation). Naturally, such models will need to account for and accurately predict all types of variation whether they occur upstream, downstream, or at the specific gene of interest. These models do so either by introducing additional states (i.e., changing N) or by introducing hyper-parameters to capture variability in the rates (i.e., by adding extrinsic noise). It might be interesting to explore if the authors' work can shed light on when each of these approaches is more or less likely to succeed to create a well-constrained model capable to reproduce and predict experimental results.

In the results shown in Fig 2, the authors description of their parameter sampling is insufficient. It is not clear in the main text what was meant by the statement, "Thousands of rate parameter sets of the N -state model ($N = 3, 4, 5$) were sampled in the range relevant for eukaryotic gene expression." The SI shows that the authors are sampling from uncorrelated log-uniform, so it is not surprising that most parameter sets would result in a case where one is much smaller than the rest, and there is a single rate-limiting set between successive returns to the ON state. On the other hand, if the sampling involved strongly correlated parameters (i.e., where all forward transition rates had similar scales), then I expect that you will get many more instances of models displaying the more complex phenomena.

The statement, "The similarity of the N -state and effective telegraph model means that the maximum likelihood of each distribution computed using mRNA count data from each cell would also be similar and hence using common model selection criteria such as the Akaike Information Criterion or the Bayesian Information Criterion (which penalise more strongly models with a larger number of parameters), the telegraph model would be selected as the optimal one," needs more support. For very large numbers of measurements, as the authors assume at many points throughout the manuscript, I would expect the AIC and BIC approaches to always select the most complex model. It would be more compelling if the authors directly estimated the average AIC and BIC versus numbers of cells for data generated from certain models and show how many cells are needed before one selects the true number of states.

The statement, "Now it has been proved that when the steady-state distribution of the telegraph model is bimodal then one of its peaks must lie at zero [46]," (and indeed all analyses in the current manuscript) assumes that there is no leaky transcription from the first state. However, for many regulated processes, this will not be true. The authors later address this to state that "Note that while in principle j (the integer labelling the initial inactive state) can be variable between cells, in practice this is unlikely because the inducer used for gene induction is associated with a particular stage of the transcriptional initiation process, hence implying a fixed value of j for all cells in the population," which is not supported by citation to relevant literature and seems difficult to prove. I would expect that some cells could be poised to respond (e.g., in a state j_1 where the chromatin is accessible) while others are not poised to respond (e.g., in a state $j_2 < j_1$ that that is less advanced in the process to fully active transcription). Rather than making these unsupported claims that most gene regulation processes start in a single OFF state with zero transcription (conditions that appear necessary to allow the use of the authors' proposed approach), the authors can simply state that they are focusing their attention on a subset of GRN processes that begin in a single-initial OFF state. In addition to making the focus of the paper more clear, explicitly stating such a focus would provide a more rigorous rationale for their decision to focus on specific genes in Fig 5E-H in the section where they state "...we chose six induction curves (for the genes *Tnfaip3*, *Icam1*, *Cxcl1*, and *IL-1b*) whose initial RNA count was approximately 0 and for which the mRNA curves lagged behind the pre-mRNA curves by several minutes."

The authors spread their attention too thin in trying to describe steady state analyses at the beginning of the paper and temporal power-law analysis at the end. As a result, the paper becomes less focused and does not provide a sufficiently thorough investigation of the practical use of the power-law behavior. It would have been preferable to move the steady state analyses to the SI, so that the authors could more thoroughly explore the power-law in the main text.

The parameters used in Fig 3 appear to be very similar for the k_i values. How were these chosen? It seems unlikely that they were drawn by chance from the broad distributions (log uniform in $[0.1, 40]$) described previously, since these would lead to less uniform differences between the results for the different starting values for j .

In Fig 4, the reason why the authors required the cells to reach a mean of 10 at a time of 0.1/d is not clear to me. I was curious how the expression at $t = 0.1$ would relate to the actual steady state number (MSS). For a 1-state process, this is calculated easily: $M(0.1)/MSS = 1 - \exp(-0.1) = 0.095$ meaning that the SS number would be ~ 100 , which seems reasonable. But for a 3-state process, e.g. where $k_1=k_2=k_3=1$ and $\rho = 1$, if $M(0.1)=10$, then the MSS needs to be about 65,000 and for $N = 4$, then MSS would need to be 2.6×10^6 . Such numbers are not reasonable, and I would urge the authors to either drop the requirement that the mean expression reaches 10 before the time 0.1, or do a better job to explain the meaning of this constraint.

In the discussion, I was confused by the section "We note that while an extension of the telegraph model to include transcription from both the active and inactive states (the so-called leaky telegraph model [58]) would predict a qualitatively similar distribution shape (in the limit of slow switching between active and inactive states), there still remains an interpretation issue. This is because the notion of a leaky off-state implies low expression and hence a peak which is very close to zero – but as is clear from Fig. 2G, both peaks can be far away from zero – hence in this case both the telegraph model and its leaky variant can be safely excluded." I do not see why the leaky telegraph would need to have a nearly-zero rate of transcription in the lower activity state. Why not just have two states with two rates that are both nonzero?

The following statements are not clear and need further explanation or support, "As for nuclear export, the difference in the exponents of the power laws fitted to spliced and unspliced data excludes the possibility of a step with fixed time delay to describe splicing." and "We found no evidence that nuclear export or splicing can be described by an effective reaction step with a fixed time delay – rather these seem to be described by one rate-limiting step with first-order kinetics."

(Remarks on code availability)

Version 1:

Reviewer comments:

Reviewer #1

(Remarks to the Author)

The authors have addressed successfully the comments raised by all the referees during the review process.

(Remarks on code availability)
(same as previous review)

Reviewer #3

(Remarks to the Author)

Second review for "Transient power-law behaviour following induction distinguishes between competing models of stochastic gene expression"

I found the authors' responses and revisions to be thorough and convincing, and the resulting manuscript is much improved. I am still of the opinion that the steady state analyses in the first half of the manuscript is unsurprising and could be shortened, but the second section is nicely expanded making the manuscript as a whole an interesting and valuable contribution to the field. Overall, I am satisfied with the modifications.

I had only one remaining point of concern from the authors' response and revision, specifically about the authors' response regarding the starting of the cells in a distribution of initial states:

Author Response Text – "Regarding assumption (2) we think it is unlikely there is a wide distribution of initial states. If there was such a distribution then theory would predict that that the mean count increases with time via not a power law but a sum of terms each of which is a different power law. This would mean that it would not be possible to fit a simple power law to the induction data — the simple fact that the yeast and mouse data follow such a simple law does suggest that the distribution of initial states is narrow."

Relevant text in Manuscript, Page 10 – "Hence, the distinguishing feature of a wide distribution of initial states is that it will not be possible to fit a single power law to the short-time data."

This is an important result that helps define the scope of the authors' approach. The explanation sounds reasonable, but I lack sufficient intuition to know how easy it would be to distinguish the mixture of multiple power law curves from that of a single power-law or how many cells would be needed to discern that difference using just 3 or 4 time points. So, I would like to see a little more evidence to back up this claim. Specifically, it would be nice to see a relatively simple study (and another two SI figures) as follows:

(1) Consider an example where the cells start in some non-delta distribution of initial states along the TS activation chain as follows. Assume that there are $N-1$ inactive states in a linear chain with forward and backward transition rates chosen as before and a single active N th state that can be reached from the $(N-1)$ th state only after an inducer has been applied.

(2) Start the process at the equilibrium distribution for the first (N-1) states (or just run the SSA starting at a large negative time with $t=0$ being the time at which the inducer is added).

(3) Show that (at least for some parameter combination) the mean expression of this system after application of induction CANNOT be fit with an initial power law allowing one to rigorously reject the hypothesis of a single starting point using an experimentally feasible amount of data. This plot would be an interesting negative control.

(4) Then, after showing that this is the case for one particular hand-chosen parameter set, do another sweep over parameter sets with this new model. For each parameter set, classify them as being well or poorly fit by the power law induction curve, and show that, "the distinguishing feature of a wide distribution of initial states is that it will not be possible to fit a single power law to the short-time data." Perhaps you could plot the power law fitting error versus the entropy of the initial distribution versus, where a strong correlation with a steep slope would provide a clear demonstration of this result.

(Remarks on code availability)

Version 2:

Reviewer comments:

Reviewer #3

(Remarks to the Author)

After reading the authors' thorough response, I am satisfied with the new version of the manuscript, and I have no further comments or suggestions.

(Remarks on code availability)

Response to reviewer comments

We thank all reviewers for their detailed comments. We have endeavoured to address all of them, which has significantly improved the manuscript. The response to the comments are in blue and the associated changes in the manuscript are marked in red.

Reviewer 1

In the manuscript the authors study the problem of the identifiability of the structure of transcriptional states from mRNA count data. On one hand the authors show that steady state distributions of mRNA a classic telegraph model is virtually indistinguishable from more sophisticated multistate models. While this result is interesting, it is not surprising. On the other hand, I found very interesting the second part of the paper, where they show elegantly how time course data of average mRNA number of a gene being induced can be used to extract information about the underlying number of states. Overall the paper is interesting, relevant, well written, and thoroughly documented.

Some more detailed comments/suggestions:

1) The authors made clear (using relevant experimental references) that parameters used along the manuscript are based on relevant ranges. Nevertheless, it is not clear how some of the cases were chosen. For instance the parameters used for Fig.3 A-D. Is there any rationale behind this choice?

The parameters for Fig. 3 (shown in SI Table III) were chosen to be within the biological range used to generate Fig 2 (SI Note 5.1). We have added a sentence to the caption of Fig. 3 to clarify this point. We note that this is only intended to be an illustrative example — the power law results given by Eqs. (3)-(4) exist independent of the actual parameter values used.

2) I did not fully understand the scheme of panel F t^4 . Is the scheme a state in which the successful production of mRNA returns the system to state 2 with rate $\rho = 1600$, or it returns to the state 1 with rate $k_5 = 0.5$? If this is the case, aren't the rate to each other, and the result would be the same as not having the 1st state? Rate $k_{21} = 0.5$ seems also to be very slow.

The scheme associated with the power law t^4 describes a system where after induction the system is in state $j = 2$ and then it can reach the active state ($j = 5$) via a number of different transitions. When in the active state, the system can move to the inactive state $j = 1$ without any mRNA production event or else an mRNA is produced and simultaneously the state changes to $j = 2$; see reaction scheme (90) in SI Note 5.2. The size of the rate constants is unimportant since for all possible rates we prove in SI Note 4 that the power law is t^4 for short times because the shortest path from $j = 2$ to a newly formed transcript consists of 4 steps. We have added a sentence to the caption of Fig. 3 referring the reader to SI Note 5.2 for reaction network schemes.

3) One of the experimental problems with mRNA counting is the detection accuracy. This results in the observations of only a fraction of the mRNAs being counted. How would this affect the results of the theory? As a first approximation I think the theory should hold. Assuming that the average mRNA number measured is a fraction of the real number of mRNA, it would preserved the exponent of the power law. On the other hand, due to many cells with low numbers of mRNA at early time points will bias this result.

If we have an infinite number of cells and only a fraction f of the mRNAs in each cell are counted this implies that the proportionality constant in Eq. (3) is multiplied by f but the power law remains unchanged. Hence the use of the method with bulk sequencing data presents no problem since in this case the number of cells is very large (tens of thousands to millions). For

experiments with single-cell technologies (scRNA-seq or sm-FISH) the number of cells is much less (few hundreds to few thousands) and hence in this case it is less clear how the detection accuracy will impact the theoretical results. To understand this scenario, we have used the following simulation protocol.

We first chose a subset of 20 different parameter sets from those previously analyzed in Fig. 4. These are distinguished from each other by the mean mRNA count at the last time point which varies between 1 and 25 to capture the realistic range of mRNA abundance in mammalian cells (the median mRNA count per cell for a population of exponentially growing mouse fibroblasts is approximately 17). For each parameter set, we used the SSA to simulate the mRNA count in each cell of a finite population of 1000 cells for a short period after induction. The observed mRNA count in a cell with n counts is subsequently obtained by drawing a binomial random number with success probability p and number of trials equals to n . Note that the success probability is a measure of detection accuracy. From this we calculate the mean mRNA count versus time curves sampled at 5 time points and we estimate the exponent of the power law (Eq. 3) using regression. Repeating this procedure 5000 times leads to a distribution of exponent values for each parameter set and each value of detection accuracy ($p = 0.2, 0.4, 0.6, 0.8, 1$). These are shown by boxplots in Figs. S6-S9 for the N -state model with $N = 5, j = 1$ and the N -state model with $N = 5, j = 4$ using both linear and non-linear regression. We find that the exponent of the power law is practically independent of the success probability p even when the mean mRNA was as low as 1 at the final time point. Of course, even if the success probability is very high, we expect significant errors in the exponent will start to appear if the number of cells is very small (of the order of a 100 — see Fig. S2-S5) and in this case the method is not applicable. These results are now discussed on P. 11 paragraphs 4-7 of the main text.

4) The power law fits (Fig. 4) only allow to predict a lower bound for the number of states from the initial state of the gene. Nevertheless, Fig. 4 seems to confirm that it is not any lower bound, but actually a good predictor of the number of states. For almost all the cases the next integer to the float number of the predicted slope is the correct answer. I think this is a strong result! Do the authors predict that this result will hold in real situations?

Unfortunately this is not the case. In Fig. 4 we show that in the limit of infinite number of cells, the estimated exponent from the mean count versus time data, collected over a realistic time range for mammalian cells, is a lower bound to the theoretical prediction of the exponent in the limit of short times after induction (red vertical dashed line). For clarity we have now added in brackets $N - j + 1$ in the legend so that it is clear that the line does not show the number of gene states N . For example, in Fig. 4H the true number of states is 5 but the theoretical exponent is 2 because the initial state is $j = 4$ and in fact the distribution of the estimated exponent has most of its probability mass in the range 1 to 2.

5) The power law fits for real data in Fig.5 do not use a log-log scale as employed in the rest of the manuscript. I suspect that the authors chose this visualization because the measured number of mRNA is too low (or even zero) at early timepoints, resulting in large deviations from the fit in the log-log plot. How did the fitting took this into account? If the fitting was done in the linear scale, that means that initial points were not used to fit the slope of the power law, so effectively all the information of the fitting is concentrated in the last 3 timepoints. Related to this, how was the error calculated for the exponents in this fitting? I find that clarity in this section is relevant, since it illustrates the applicability of the method.

Previously for simulated data (Fig. 4) we obtained the exponent in the power law from the slope of a log-log plot of mean mRNA counts vs time. In contrast, for experimental data we obtained the exponent by directly using non-linear regression on the mean mRNA vs time data. In the new version of the manuscript, we have compared the two types of fitting using simulated data

— comparing Fig. S2 with Fig. S3 and Fig. S4 with Fig. S5 we see that the results are very similar for the N -state model with $N = 5, j = 4$ and $N = 5, j = 1$, respectively. Any small differences between the two methods become negligible when the number of cells is significantly larger than a hundred cells. This conclusion remains the same when detection efficiency is accounted for (Fig. S6-S9). Now these fits using simulated data counted time from the point at which transcription initiation starts in the nucleus, i.e. it assumed that the delay from the application of a stimulus to the binding of some transcription factor to the DNA is known. They also assumed that the mRNA count when initiation starts is zero. Generally these assumptions may not reflect the actual experiment. For the experimental eukaryotic data that we considered (yeast and mouse) the delay is unknown and the mRNA count when initiation starts appears to be small but non-zero. In the new version of the manuscript we show (SI Note 4) that if we take into account these subtleties, the equation for the mean mRNA count for short times is modified to

$$\langle m(t) \rangle - \langle m(0) \rangle e^{-dt} = \begin{cases} 0 & t < t_0, \\ \frac{A}{n!} (t - t_0)^n + O(t^{n+1}) & t \geq t_0, \end{cases} \quad (1)$$

where t_0 is the delay time, $\langle m(0) \rangle$ is the mean mRNA at time 0, i.e. when the stimulus is first applied and d is the decay rate of the mRNA. This equation cannot be reduced to the form of a straight line in log-log space and hence there is no option but to estimate the exponent n (and the other parameters) using non-linear regression. If single-cell data is available then the regression can be performed on each bootstrapped sample of the data thus giving the sampling distributions for the estimated parameters. An application of this type is shown in the new Fig. 5 for yeast data. We note that the estimation of the exponent is largely unaffected by setting $d = 0$ because the mRNA decay rates in yeast are quite small with a median value of merely $d = 0.06/\text{min}$ (corresponding to a median half life of 11 mins). However the estimation is considerably affected (exponent is overestimated; Fig. S12) if the delay is ignored (set to zero) because the delay time is approximately 3 minutes which is comparable to the time range over which the power law is fit. Note that we do not use all datapoints for the fitting; the optimal number of points is selected by an automated procedure based on the residual sum of squares (Fig. S10 and SI Note 6.4). This more detailed analysis (explained in the manuscript main text on pages 12-13) clarifies the applicability of the method to real data. Because the mouse data is not single-cell, we can only fit Eq. (1) to the population mean count vs time data (Fig. S13). Thus the uncertainty in the exponents and other parameters here reflects the scatter of the data about the best fit line (Table S7 and SI note 6.6).

(6) The code is presented in a Jupyter notebook using Julia written and commented clearly. The output plots follow the same style as the ones shown in the paper making very easy to reproduce, use, and recycle the code.

We thank the reviewer for the positive comment.

Reviewer 2

(1) Briefly, the authors present a linear model for gene transcription (with many linear gene OFF states culminating in an ON state) and present analytical expressions for the average number of mRNA as a function of time. From the theory of first passage times (<https://arxiv.org/pdf/1503.00291.pdf>) on a linear set of states, we can compute the mean level mRNA in time.

Intuitively, we know that the more OFF states it takes to get from transcription initiation, through all OFF states, to gene product output (successful transcription of one mRNA), the longer it takes on average for the first gene product to appear.

This is quantified by the theory of first passage times where the mean level of mRNA (or any analyte) is expected to be power law initially.

This observation is used in all applications where the number of states are estimated from early times (e.g., see Figs. 1-2 from <https://www.science.org/doi/10.1126/sciadv.aaz4642> but there are many more examples scattered across the literature). In principle this observation has been used since single ion patch clamp (in the 80's) where the number of non-conducting states were to be estimated.

While the work highlights an application of a well-known linear chain model, it really does not rise to the level of novelty expected here for this referee.

We thank the Referee for pointing out the related paper in Science Advances. This paper considers a continuous-time Markov process that is related to ours, and establishes an early-time power law in the probability distribution of the first passage time. Both this paper, and earlier theoretical papers by Li, Kolomeisky and Valleriani (The Journal of Chemical Physics 139.14 (2013); The Journal of Physical Chemistry B 118.35 (2014): 10419-10425; The Journal of Chemical physics 140.6 (2014)) are relevant to our work, and we have included them in the revised version of the manuscript by a brief discussion when the power law result is first introduced (page 9 of the main text). We note that besides these papers we have also cited Journal of Applied Probability 33.2 (1996): 368-381. From theorem 3.1 of this paper, one can derive the same power law results as the papers mentioned above. However, we note that the measurement of the first passage time distribution from induction to the first mRNA transcript is difficult to achieve in experiments probing stochastic gene expression; rather it is the mean number of mRNA that is much easier to measure and that is routinely reported using single-cell technologies or bulk sequencing. We are not aware of a derivation of the power law in the mean mRNA for short times — as earlier mentioned, previous papers focused on the first-passage time distribution. In this regard we note that because we have a process with degradation, it is not *a priori* obvious that the short-time power law in the mean mRNA is the same as that obtained from the first term in the Taylor series expansion of the distribution of the first passage time to the first mRNA production event. We have discussed some of these points on P. 9 last paragraph.

Furthermore we note that as explained in the new version of the paper, explicitly accounting for the time delay from the application of the stimulus to the transcription factor binding to the DNA and initiating transcription leads to an important modification of the power law result (see also SI note 4):

$$\langle m(t) \rangle - \langle m(0) \rangle e^{-dt} = \begin{cases} 0 & t < t_0, \\ \frac{A}{n!} (t - t_0)^n + O(t^{n+1}) & t \geq t_0, \end{cases}$$

where t_0 is the delay time, $\langle m(0) \rangle$ is the mean mRNA at time 0, i.e. when the stimulus is first applied and d is the degradation rate of the mRNA. We note that this result is not just valid for the linear chain model but for a wide class of systems of interest in gene expression whose

reaction scheme is given by Eq. (5) in the main text. We find that the use of this equation rather than the simple power law is crucial to the correct estimation of the exponent and hence of the number of gene states. Ignoring this time delay leads to a considerable overestimation of the exponent (Fig. S12). This is now discussed in the manuscript main text on pages 12-13. We emphasize that a main message of our paper is that the application of this result to widely available transcriptomics data is much simpler than commonly used sophisticated approaches based on maximum likelihood or Bayesian inference. Within the specific field of transcriptomics, the method proposed is novel and we believe it stands on its own.

Reviewer 3

(1) In the manuscript, “Transient power-law behaviour following induction distinguishes between competing models of stochastic gene expression”, authors Nicoll et al describe a simple analysis of the mean expression of gene activity in the initial time following application of an induction signal. The authors argue that if (1) all cells start in a fixed initial state and (2) there is only one fully active transcription state, then (at least for a short time following stimulus) the mean expression level will follow a power-law behavior. Moreover, the exponent of this power-law will provide an easily calculated lower bound on the number of steps between the initial state and the transcriptionally active state.

Overall, this is an interesting paper, and I can certainly see value in having simple means to infer models for gene expression dynamics. However, I have several concerns about the manuscript in its current form as follows:

We thank the reviewer for the positive comments.

(2) The manuscript lacks sufficient discussion of uncertainty quantification in their analyses of experimental data. Without these details and a thorough quantification of posterior parameter uncertainties after fitting, it is difficult to determine if the author’s claims that they have identified the number of states is sufficiently well supported.

The statements, “Consider an experimental setup whereby some gene of interest is inactive before induction by an inducer molecule. From application of the inducer to its binding to DNA, there is some time lag due to nuclear translocation and other processes. Hence we shall consider $t = 0$ to be the point at which induction starts in the nucleus.” seems to assume that one can precisely know the time at which the inducer reaches the gene of interest. But this can be a complicated process. For example, in the data analyzed in Fig 5A-D, gene expression is induced by a time varying MAPK signal that goes through multiple phosphorylation steps before reaching the nucleus. The timing of this process could be unknown (and is possibly different for every cell), but appears to be quite fast for the case of the yeast osmotic shock. It is not clear from the manuscript (1) how sensitive is the presented analysis to errors in the time $t = 0$, or (2) how the presented analysis could be modified to account for this deviation in this time. The authors should analyze how errors in this specific time would affect their determination of the minimum number of states.

We agree with the reviewer that this is an important point. The paragraph mentioned by the reviewer, where we discuss the experimental setup, is specifically for the section where we show how well can the exponent be estimated from simulated data (Fig 4). Indeed in this case, for simplicity, time is measured from the point at which transcription initiation starts in the nucleus, i.e. it assumed that the delay from application of an inducer to the binding of some transcription factor to the DNA is known. We also assumed that the mRNA count when initiation starts is zero. Generally these two assumptions may not reflect the actual experiment. We have now clarified on P. 7 last paragraph of the main text that this simplified idealised setup makes these two assumptions.

After the discussion of Fig. 4, we mention that when fitting the model to data these assumptions may not hold and hence we extend the model to account for this more realistic scenario. In SI Note 4 we show that the equation for the mean mRNA count for short times is now modified to

$$\langle m(t) \rangle - \langle m(0) \rangle e^{-dt} = \begin{cases} 0 & t < t_0, \\ \frac{A}{n!} (t - t_0)^n + O(t^{n+1}) & t \geq t_0, \end{cases}$$

where t_0 is the delay time, $\langle m(0) \rangle$ is the mean mRNA at time 0, i.e. when the stimulus is first applied and d is the degradation rate of the mRNA. This equation cannot be brought to the form

of a straight line in log-log space and hence there is no option but to estimate the exponent n (and the other parameters) using non-linear regression. If single-cell data is available then the regression can be performed on each bootstrapped sample of the data thus giving the sampling distributions for the estimated parameters. An application of this type is shown in the new Fig. 5 for yeast data. We note that the estimation of the exponent is largely unaffected by setting $d = 0$ because the mRNA decay rates in yeast are quite small with a median value of merely $d = 0.06/\text{min}$ (corresponding to a median half life of 11 mins). However the estimation is considerably affected (exponent is overestimated; Fig. S12) if the delay is ignored (set to zero) because the delay time is approximately 3 minutes which is comparable to the time range over which the power law is fit. Note that we do not use all datapoints for the fitting; the optimal number of points is selected by an automated procedure based on the residual sum of squares (Fig. S10 and SI note 6.4). This more detailed analysis (explained in the manuscript main text on pages 12-13) shows how the method can be extended to include the time delay, i.e. the deviation from the time $t = 0$ of the simple power law (Eq. (3)), and also presents methods to estimate this delay simultaneously with the exponent.

(3) I am concerned about the practical aspects of fitting such small values of mean expression. Several plots in Figs 3, 4A, 5 show mean expression levels that are much less than one, and Fig 3 shows corresponding variances of equal magnitude. Unfortunately, this suggests situations where the standard deviations are much greater than the mean. As a result, it might be difficult to determine the mean of the mRNA to any degree of accuracy without measuring a very large number of cells. Moreover, if there is any error in those measurements (e.g., an erroneous spot call in a smFISH image processing step for a small fraction of cells), then those errors could have a major impact on the inferred exponent. Some effort needs to be taken to explore the uncertainty in these values (e.g., add error bars to the curves in Figs 4, 5) and propagate that uncertainty to compute confidence intervals for the identified exponents.

In referring to Fig 4A, the statement, "Note that by the use of these equations, here we made the implicit assumption that the number of cells is infinite (in practice this means the number of cells is larger than a few thousand such that the standard error of the mean is very small)" is not sufficiently clear and requires additional analysis. At least for the representative example depicted in Fig 4A, it would be nicer to propose a reasonable number of cells (e.g., 50, 100, 1000, 10000 per time point), generate the simulated data to find the mean, and then infer their model parameters from that simulated data. At a minimum, the authors could use bootstrapping to then determine the uncertainty of their parameter estimates.

In response to the referee's point, we have undertaken an extensive simulation-based analysis to study the effects of small mean expression levels, detection accuracy, sample size (number of cells) and fitting method (linear vs non-linear regression) on the values of the inferred exponent. For all these cases we consider the simplified scenario where the time delay from the stimulus to transcription initiation in the nucleus is known and where the initial amount of mRNA is zero.

We first chose a subset of 20 different parameter sets from those previously analyzed in Fig. 4. These are distinguished from each other by the mean mRNA count at the last time point which varies between 1 and 25 to capture the realistic range of mRNA abundance in mammalian cells (the median mRNA count per cell for a population of exponentially growing mouse fibroblasts is approximately 17). For each parameter set, we used the stochastic simulation algorithm (SSA) to generate a dataset of mean versus time (sampled at 5 time points) for a finite population of M cells and linear regression was used to obtain the exponent from the log-log plots, as before. Repeating this procedure 5000 times leads to a distribution of exponent values. In Fig. S2 we show these distributions by means of boxplots for the N -state model with $N = 5, j = 4$ and for populations of sizes $M = 100, 500, 1000$ and 10000. The medians of all boxplots are

in good agreement with the exponent estimated assuming an infinite population size (Fig. 4). Except for populations of 100 cells, the statistical dispersion as measured by the interquartile range is small indicating that for sample sizes of 500 or more cells, uncertainty in the exponent estimates due to finite sample size effects are not important. Note that this conclusion is valid independent of the mean mRNA count at the last time point, i.e. the method is robust even if the mean count is of order 1. Similar results are obtained if the inference is repeated using non-linear regression applied to the mean count versus time plots (Fig. S3) and for the N -state model with $N = 5, j = 1$ (Figs. S4 and S5).

To address the referee's comment on detection accuracy, we repeated the analysis for the case of non-perfect capture of mRNAs by single-cell mRNA measurements using smFISH or scRNA-seq. To understand this scenario, given an integer mRNA count n for a cell at a time point from the stochastic simulation algorithm (SSA), we generated the corresponding observed mRNA count by drawing a binomial random number with success probability p and number of trials equals to n . Note that the success probability is a measure of detection efficiency — it is estimated to be over 80% for FISH experiments and between 5 – 32% for single-cell sequencing experiments. In Figs. S6-S9 we show that for a typical sample size of 1000 cells, the estimated exponent obtained by linear or non-linear regression from simulated data for the N -state model with $N = 5, j = 1$ and $N = 5, j = 4$ is practically independent of the success probability p and is similar to that estimated for an infinite number of cells with perfect detection efficiency in Fig. 4.

In summary, we make two crucial assumptions of our idealised experimental setup, i.e. the time delay for the stimulus to reach the nucleus is known or can be separately measured, and the mean mRNA prior to induction is very small. If additionally the sample size is at least several hundreds of cells large, the inferred exponent (and hence a lowerbound on the number of gene states) is accurately inferred by means of linear or non-linear regression. This conclusion holds even if the mean expression levels at the last time point are very low or if the measurement method has low detection efficiency.

These points are now discussed on P. 11 paragraphs 4-7 of the main text.

(4) In the application of the approach to the data in Fig 5, there is insufficient discussion of the uncertainty in the data and how that affects parameter estimates in the resulting model. In Fig 5, I would expect to see error bars on all data points. For the data in panels A-D, these could be SEM calculated from the measured single-cell distributions at those times. For the data in E-H, these could include error bars showing the SEM between the different experimental replicas. For Table 1, the description of the approach to quantify the SEM in the parameter estimates does not provide sufficient detail. SI Note 5 only states that the authors used a nonlinear least square fitting routine to fit the measured mean at each time. This description does not include details for how the authors propagate uncertainty in the data during their fitting procedure. The authors need to provide some sort of cross-validation, bootstrapping analysis, or Bayesian posterior quantification to compute confidence intervals or uncertainties in their estimates, given the data.

As described in the response to point 2 we have redone the non-linear regression fitting using a new model that takes into account both a fixed (unknown) time delay between stimulus and transcription initiation and non-zero initial amount of mRNA. For the case of single-cell yeast data, the uncertainty in the parameter estimates is obtained by fitting to each bootstrapped mean count vs time dataset. In Fig. 5 and Fig. S11 we show the distributions of all estimated parameters. On P. 13 and Fig. S12 we discuss in detail how the uncertainty in the time delay affects the exponent.

For the mouse data this is not possible because it is not single-cell, and hence we can only fit the model to the population mean count vs time data (Fig. S13). The uncertainty in the exponents and other parameters here reflect the scatter of the data about the best fit line (SI Table 7 and SI Note 6.6).

In the new Fig 5 which focuses on the yeast data we have added SEM calculated from the measured single-cell distributions on all data points in the timecourse curves shown in C-F (nuclear data) and K-N (cytoplasmic data) — because of the significant sample size, these are quite small. For the mouse data (Fig. S13) this is not possible because the data is not single-cell and as well we do not have available experimental replicates, hence cannot display error bars showing the SEM between the different experimental replicas.

(5) Considering the issues above, the authors should provide a more detailed discussion of the limitations of their approach. For example, their analysis focuses on a limited case where: (1) the cells start with zero mRNA; (2) cells start in a single fixed state; (3) the process starts at a precisely known initial time; and (4) the mean number of mRNA is measured without measurement noise. It is not clear if/how the presented analysis could be extended to allow for non-trivial initial probability distributions for the states, uncertain initiation times, non-zero initial expression levels, or measurement errors in the quantification of expression. Although it may be unreasonable to tackle all these concerns in a single paper, each of these major assumptions should either be specifically addressed in the manuscript or described at the end as a current limitation that requires future investigation.

As described in the response to points 2 above, the new model used to fit the experimental data does not make assumptions (1) and (3). Assumption (4) we tested using simulated data as described in the response to point 3 above and found that as long as the number of cells is larger than several hundred cells (a common situation) then measurement noise is not an issue. Regarding assumption (2) we think it is unlikely there is a wide distribution of initial states. If there was such a distribution then theory would predict that the mean count increases with time via not a power law but a sum of terms each of which is a different power law. This would mean that it would not be possible to fit a simple power law to the induction data — the simple fact that the yeast and mouse data follow such a simple law does suggest that the distribution of initial states is narrow. We have now discussed these points on P. 10 paragraph 2 and P. 13 paragraph 3.

(6) Beyond the technical aspects of determining N from data using early time dynamics, it is not clear from the manuscript what is the utility for determining this number of states. Since the concept of discrete states is a convenient simplification of a much more complicated reality, it would be nice if the manuscript could provide more insight into questions such as: What would one do with the lower bound N once it is determined? How would the knowledge of N this assist the researcher to build more predictive models, design more effective experiments, or discriminate more effectively between competing hypotheses for regulation mechanisms?

One may ask what is the practical utility of determining a lower bound on the number of gene states using the method that we have here developed. Transcriptional regulation involves several distinct steps, and it remains unclear which of these are targets of regulation. As noted earlier, the number of inactive states can be seen as being equal to the number of rate-limiting reaction steps in initiation which are likely the key control points of transcriptional regulation. The method described allows one to rapidly estimate a lower bound for the number of these points. If the experiment is repeated with different perturbations then one can possibly establish a biological interpretation for each of the discrete gene states. So for example, say we wanted to understand the rate-limiting steps in mammalian transcription. *In vitro* experiments have identified six transcription factors (TFIID, TFIIA, TFIIIB, TFIIIF, TFIIIE and TFIIH) that must

bind in a certain order before the preinitiation complex begins elongation. The fact that the number of essential transcription factors is significantly larger than the number of states of models commonly fit to mammalian gene expression data suggests that only the binding of a few transcription factors is rate-limiting. We can then set up a series of induction experiments which differ from each other only by the overexpression of one transcription factor. In all of these experiments we assume the same inducer is used such that the distribution of the initial gene state remains narrow and centered on the same value. If a transcription factor is under normal conditions associated with a rate-limiting step, when it is over expressed it will not be anymore since the reaction will be speeded up — in this case, we expect the number of gene states to decrease by one (compared to that in normal conditions) which will result in a similar decrease in the exponent of the power law estimated from the short-time kinetics. We have now discussed these points in the penultimate paragraph of the Discussion section.

Minor Concerns:

I disagree with the statement, "Despite this detailed molecular knowledge, it has thus far proved impossible to quantitatively predict the kinetics of gene expression at the single-cell level." Many of the papers cited have accurately predicted the distributions of single-cell gene expression upon fitting very similar types of models to data.

We have removed this sentence from the first paragraph of the Introduction and instead have written: "Despite this detailed molecular knowledge, it has thus far proved difficult to understand the large degree of heterogeneity in gene expression observed in a population of cells."

The sentence in the introduction, "We note that while an extension of this model to include reversible reactions between gene states or transcription from multiple gene states is likely more realistic [17–19], nevertheless it has many more rate parameters than the irreversible model and hence the latter is typically preferred," needs more support for the second clause. Most of the models I have seen that have fit and predicted single-cell data have included reversible reactions, including the ones cited in this specific sentence. Since the authors explore this case later in the manuscript, it would be worth mentioning that this approximation is relaxed later in the paper.

We have rewritten the sentence to read: "We note that while an extension of this model to include reversible reactions between gene states or transcription from multiple gene states is likely more realistic (refs) (and we shall consider it later on), nevertheless it has many more rate parameters than the irreversible model and hence the latter is in common usage (refs)." We also cited two references to show that the irreversible model is commonly used. This can be found on Pages 1-2.

The discussion of the interplay between varying numbers of gene states and the effect of extrinsic noise is not sufficiently clear. The authors describe previous efforts toward finding the optimal number of states, N , needed to capture and predict observed data. In such problems, the goal is to find the model that is best supported by available data (e.g., through analyses using cross-validation or Bayesian estimation). Naturally, such models will need to account for and accurately predict all types of variation whether they occur upstream, downstream, or at the specific gene of interest. These models do so either by introducing additional states (i.e., changing N) or by introducing hyper-parameters to capture variability in the rates (i.e., by adding extrinsic noise). It might be interesting to explore if the authors' work can shed light on when each of these approaches is more or less likely to succeed to create a well-constrained model capable to reproduce and predict experimental results.

This is an interesting suggestion. In Fig. 3E in the previous (and current) version of the manuscript, we showed that static extrinsic noise does not influence the exponent of the power

law and hence the estimate of the lowerbound of the number of gene states, i.e. as far as the short-time dynamics are concerned there is no interplay between varying the numbers of gene states and the effect of extrinsic noise. Hence the exponent of the power law determines the number of gene states but cannot be used to say anything about extrinsic noise. This is also discussed on P. 10 paragraph 2.

In the results shown in Fig 2, the authors description of their parameter sampling is insufficient. It is not clear in the main text what was meant by the statement, "Thousands of rate parameter sets of the N -state model ($N = 3, 4, 5$) were sampled in the range relevant for eukaryotic gene expression." The SI shows that the authors are sampling from uncorrelated log-uniform, so it is not surprising that most parameter sets would result in a case where one is much smaller than the rest, and there is a single rate-limiting set between successive returns to the ON state. On the other hand, if the sampling involved strongly correlated parameters (i.e., where all forward transition rates had similar scales), then I expect that you will get many more instances of models displaying the more complex phenomena.

This is a reasonable hypothesis and hence we tested it. In SI Fig. 14 (see also SI Note 6.7) we verify that this is not the case: mRNA count distributions from N -state models with $N > 2$ and $k_i = a$ for $i = 1, \dots, N - 1$ where a is some constant are well fit by an effective telegraph model. This strengthens our earlier conclusion that for shapes I-III, steady-state data is not sufficient to distinguish between N -state models with $N = 2, 3, 4$ or 5 . We now discuss this briefly on P. 7 first paragraph of the main text.

The statement, "The similarity of the N -state and effective telegraph model means that the maximum likelihood of each distribution computed using mRNA count data from each cell would also be similar and hence using common model selection criteria such as the Akaike Information Criterion or the Bayesian Information Criterion (which penalise more strongly models with a larger number of parameters), the telegraph model would be selected as the optimal one," needs more support. For very large numbers of measurements, as the authors assume at many points throughout the manuscript, I would expect the AIC and BIC approaches to always select the most complex model. It would be more compelling if the authors directly estimated the average AIC and BIC versus numbers of cells for data generated from certain models and show how many cells are needed before one selects the true number of states.

We agree with the reviewer that in the limit of infinite number of cells, one could separate the N -state and effective telegraph model from steady-state measurements. However for finite and realistic sample sizes this is not the case. For example in the paper Jiao, Feng, et al. "What can we learn when fitting a simple telegraph model to a complex gene expression model?" PLOS Computational Biology 20.5 (2024): e1012118 they used maximum likelihood to fit the telegraph model and the three-state model to steady-state mRNA count data generated from stochastic simulation algorithm (SSA) simulations of the three-state model. Model selection was then performed using the corrected Akaike information criterion (AICc). They showed that this leads to incorrect model selection (telegraph model was selected) for over 90% of parameter sets if the sample size is 100 cells (typical for scRNA-seq data), and the proportion is over 40% even for 10,000 cells (larger than for most smFISH datasets). We have discuss these points on P. 7 paragraph 2 of the revised paper.

The statement, "Now it has been proved that when the steady-state distribution of the telegraph model is bimodal then one of its peaks must lie at zero [46]," (and indeed all analyses in the current manuscript) assumes that there is no leaky transcription from the first state. However, for many regulated processes, this will not be true. The authors later address this to state that "Note that while in principle j (the integer labelling the initial inactive state) can be variable between cells, in practice this is unlikely because the inducer used for gene induction is as-

sociated with a particular stage of the transcriptional initiation process, hence implying a fixed value of j for all cells in the population,” which is not supported by citation to relevant literature and seems difficult to prove. I would expect that some cells could be poised to respond (e.g., in a state j_1 where the chromatin is accessible) while others are not poised to respond (e.g., in a state j_2 that is less advanced in the process to fully active transcription). Rather than making these unsupported claims that most gene regulation processes start in a single OFF state with zero transcription (conditions that appear necessary to allow the use of the authors’ proposed approach), the authors can simply state that they are focusing their attention on a subset of GRN processes that begin in a single-initial OFF state. In addition to making the focus of the paper more clear, explicitly stating such a focus would provide a more rigorous rationale for their decision to focus on specific genes in Fig 5E-H in the section where they state “. . . we chose six induction curves (for the genes *Tnfaip3*, *Icam1*, *Cxcl1*, and *IL-1b*) whose initial RNA count was approximately 0 and for which the mRNA curves lagged behind the pre-mRNA curves by several minutes.”

We have reworded the sentence on P. 10 “Note that while in principle etc” to read: “Note that we have here assumed that the initial state j at $t = 0$ does not vary from cell to cell. Generally given a distribution of j , say $P(j)$, then instead of Eq. (3) we have a superposition of different power laws weighted by $P(j)$. Hence, the distinguishing feature of a wide distribution of initial states is that it will not be possible to fit a single power law to the short-time data. We shall assume in what follows that this distribution is narrow and discuss this point again when we fit models to experimental data.” On P. 13 in the discussion of experimental data we have furthermore stated: “The presence of a single power law also suggests that the distribution of initial states prior to induction is narrow, an assumption that we have made throughout the paper.”

The authors spread their attention too thin in trying to describe steady state analyses at the beginning of the paper and temporal power-law analysis at the end. As a result, the paper becomes less focused and does not provide a sufficiently thorough investigation of the practical use of the power-law behavior. It would have been preferable to move the steady state analyses to the SI, so that the authors could more thoroughly explore the power-law in the main text.

We feel that there is a natural progression in our paper from firmly establishing that using steady-state data it is difficult to estimate the number of gene states to using the short-time data to accomplish this task. Hence we have left the steady-state analysis in the main text but at the same time, as clear from the responses to previous points, we have also made substantial changes which discuss more in depth the power law and its fitting to synthetic and experimental data. We believe that the narrative in the revised main text provides a more complete story than if we had to exclusively focus on the power law results.

The parameters used in Fig 3 appear to be very similar for the k_i values. How were these chosen? It seems unlikely that they were drawn by chance from the broad distributions (log uniform in $[0.1, 40]$) described previously, since these would lead to less uniform differences between the results for the different starting values for j .

The parameters for Fig. 3 (shown in SI Table III) were chosen to be within the biological range used to generate Fig 2 (SI Note 5.1). We note that this is only intended to be an illustrative example — the power law results given by Eqs. (3)-(4) exist independent of the actual parameter values used.

In Fig 4, the reason why the authors required the cells to reach a mean of 10 at a time of $0.1/d$ is not clear to me. I was curious how the expression at $t = 0.1$ would relate to the actual steady state number (MSS). For a 1-state process, this is calculated easily: $M(0.1)/MSS = 1 - \exp(-0.1) = 0.095$ meaning that the SS number would be 100, which seems reasonable.

But for a 3-state process, e.g. where $k_1 = k_2 = k_3 = 1$ and $\rho = 1$, if $M(0.1) = 10$, then the MSS needs to be about 65,000 and for $N = 4$, then MSS would need to be 2.6×10^6 . Such numbers are not reasonable, and I would urge the authors to either drop the requirement that the mean expression reaches 10 before the time 0.1, or do a better job to explain the meaning of this constraint.

We have now implemented a new parameter sampling procedure which is described in SI Note 5.3. In particular we now sample from the same biologically realistic range used for Fig. 2, we do not strictly require the mean mRNA to equal 10 (but to be at most few tens) at the last time point and the number of parameter sets is almost two million. All parameter sets used to generate Fig. 4 are consistent with a steady-state mRNA below a thousand (the median steady-state is approximately 300). However note that the timescale to reach steady-state is that of mRNA degradation, i.e. several hours in mammalian cells (for e.g. 9 is median for mouse fibroblasts), which is comparable to the cell-cycle duration (roughly 24 hours) implying that mRNA rarely reaches a steady-state for these cells — hence the steady-state calculation might not be very relevant for our purposes.

In the discussion, I was confused by the section "We note that while an extension of the telegraph model to include transcription from both the active and inactive states (the so-called leaky telegraph model [58]) would predict a qualitatively similar distribution shape (in the limit of slow switching between active and inactive states), there still remains an interpretation issue. This is because the notion of a leaky off-state implies low expression and hence a peak which is very close to zero – but as is clear from Fig. 2G, both peaks can be far away from zero – hence in this case both the telegraph model and its leaky variant can be safely excluded." I do not see why the leaky telegraph would need to have a nearly-zero rate of transcription in the lower activity state. Why not just have two states with two rates that are both nonzero?

We have rewritten the sentence in the Discussion section to read: "We note that an extension of the telegraph model to include transcription from both gene states (the so-called leaky telegraph model [ref]) can predict a qualitatively similar distribution shape (in the limit of slow switching between the two states). This implies that fitting such a model to steady-state count data characterised by distributions of shape IV may erroneously lead one to infer two active states when in reality there is only one active state. "

The following statements are not clear and need further explanation or support, "As for nuclear export, the difference in the exponents of the power laws fitted to spliced and unspliced data excludes the possibility of a step with fixed time delay to describe splicing." and "We found no evidence that nuclear export or splicing can be described by an effective reaction step with a fixed time delay – rather these seem to be described by one rate-limiting step with first-order kinetics."

In the previous version of the paper, six mRNA induction curves were fit using the simple power-law model. After refitting using the model with delay and non-zero mRNA (Eq. 6), for three of these curves the estimation gave us the non-physical result of zero delay between stimulus and the start of transcription initiation in the nucleus. This is because in these cases there are an insufficient number of measurements of the mean mRNA count to reliably estimate the delay, which then reduces the robustness of the other parameter estimates. In Fig. S13 we show three examples where the fitting was successful. Comparing the left and middle figures (pre-mRNA and mRNA of gene Cxcl1) we find the difference in exponents to be 0.82. By the results of Fig. 3 and SI Note 2.1, since the difference in exponents is close to 1 (not zero) it follows that splicing is consistent with a first-order process and not a process with deterministic delay. We have clarified these points on P. 14 paragraph 1.

Response to reviewer comments

We have undertaken a new investigation to answer the new questions of Reviewer 3. Their comments are shown in black and our response is highlighted in blue. The associated changes in the manuscript are marked in red.

Reviewer 3

Second review for “Transient power-law behaviour following induction distinguishes between competing models of stochastic gene expression”

I found the authors’ responses and revisions to be thorough and convincing, and the resulting manuscript is much improved. I am still of the opinion that the steady state analyses in the first half of the manuscript is unsurprising and could be shortened, but the second section is nicely expanded making the manuscript as a whole an interesting and valuable contribution to the field. Overall, I am satisfied with the modifications.

We thank the reviewer for their positive comments.

I had only one remaining point of concern from the authors’ response and revision, specifically about the authors’ response regarding the starting of the cells in a distribution of initial states:

Author Response Text – “Regarding assumption (2) we think it is unlikely there is a wide distribution of initial states. If there was such a distribution then theory would predict that that the mean count increases with time via not a power law but a sum of terms each of which is a different power law. This would mean that it would not be possible to fit a simple power law to the induction data — the simple fact that the yeast and mouse data follow such a simple law does suggest that the distribution of initial states is narrow.”

Relevant text in Manuscript, Page 10 – “Hence, the distinguishing feature of a wide distribution of initial states is that it will not be possible to fit a single power law to the short-time data.”

This is an important result that helps define the scope of the authors’ approach. The explanation sounds reasonable, but I lack sufficient intuition to know how easy it would be to distinguish the mixture of multiple power law curves from that of a single power-law or how many cells would be needed to discern that difference using just 3 or 4 time points. So, I would like to see a little more evidence to back up this claim. Specifically, it would be nice to see a relatively simple study (and another two SI figures) as follows:

- (1) Consider an example where the cells start in some non-delta distribution of initial states along the TS activation chain as follows. Assume that there are $N-1$ inactive states in a linear chain with forward and backward transition rates chosen as before and a single active N th state that can be reached from the $(N-1)$ th state only after an inducer has been applied.
- (2) Start the process at the equilibrium distribution for the first $(N-1)$ states (or just run the SSA starting at a large negative time with $t=0$ being the time at which the inducer is added).
- (3) Show that (at least for some parameter combination) the mean expression of this system after application of induction CANNOT be fit with an initial power law allowing one to rigorously reject the hypothesis of a single starting point using an experimentally feasible amount of data. This plot would be an interesting negative control.
- (4) Then, after showing that this is the case for one particular hand-chosen parameter set, do another sweep over parameter sets with this new model. For each parameter set, classify them as being well or poorly fit by the power law induction curve, and show that, “the distinguishing feature of a wide distribution of initial states is that it will not be possible to fit a single power law to the short-time data.” Perhaps you could plot the power law fitting error versus the entropy

of the initial distribution versus, where a strong correlation with a steep slope would provide a clear demonstration of this result.

We thank the reviewer for their insightful comment. We have performed an extensive numerical investigation to better understand the impact of non-Dirac-delta initial gene state distributions on our results. The results are explained in detail in the new SI Sections 2.2 and 6.8, Tables S9 and S10 and SI figures S15 and S16; in the main text, a summary is provided on P.10-11.

For a general initial gene state distribution, the short-time mean mRNA count is given by a weighted sum of power laws, $\langle m(t) \rangle = \sum_i A_i t^i$, where A_i are some constants, whereas for a Dirac-delta gene state distribution (all cells in one gene state initially) the mean mRNA count is given by a single power law. A log-log plot of $\sum_i A_i t^i$ versus t cannot be perfectly linear, unlike the case of a single power law. Hence, *in principle*, it is possible to distinguish a weighted sum of power laws from a single one. However, we have found that independent of the width and the type of the initial gene state distribution, the short-time dynamics of the N -state model simulated over experimentally realistic time ranges leads to mean mRNA versus time curves that are well fit by a single power law (the coefficient of determination R^2 is larger than 0.99 in most cases). Hence, *practically speaking*, it is not possible to easily determine the sharpness of the initial gene state distribution from the short-time data following gene induction. Thus, we have removed the previous claim that the distinguishing feature of a wide distribution of initial states is that it will not be possible to fit a single power law to the short-time data.

However, we emphasize that for all initial gene state distributions that we have studied, we have found that the exponent of the best fit power law still has information about the number of gene states which is the main claim of our paper. Specifically our previous claim (based on Dirac-delta initial gene state distributions) that the estimated exponent is bounded from above by the total number of gene states (N) still holds. Furthermore, we explicitly show that so long as the initial gene state distribution has a very small probability of starting in state G_{N-1} (the inactive state that is closest to the active state) then the short-time exponent is greater than 2 and hence it is straightforward to distinguish an N -state model from the two-state telegraph model. This might not be difficult to achieve because it has been shown that some repressors can force a gene into a deep off state (which would roughly correspond to one of the inactive states G_1, \dots, G_{N-2} in our model) [1]. In any case, our study clarifies how an experiment should be designed to make maximum use of the power law method.

In summary, our conclusion remains that the exponent of the best fit power law to short-time data can often be reliably used to distinguish between different models of gene expression. Our method's advantage is its simplicity and its wide applicability thus lending itself as a very useful tool in the system's biologist toolkit.

References

- [1] Aleksander T Szczurek, Emilia Dimitrova, Jessica R Kelley, Neil P Blackledge, and Robert J Klose. The polycomb system sustains promoters in a deep off state by limiting pre-initiation complex formation to counteract transcription. *Nature Cell Biology*, pages 1–12, 2024. doi: 10.1038/s41556-024-01493-w.